# Efficient Edge Test-Time Adaptation via Latent Feature Coordinate Correction

## Abstract

Edge devices face significant challenges due to limited computational resources and distribution shifts, making efficient and adaptable machine learning essential. Existing test-time adaptation (TTA) methods often rely on gradient-based optimization or batch processing, which are inherently unsuitable for resource-constrained edge scenarios due to their reliance on backpropagation and high computational demands. Gradient-free alternatives address these issues but often suffer from limited learning capacity, lack flexibility, or impose architectural constraints. To overcome these limitations, we propose a novel single-instance TTA method tailored for edge devices (TED), which employs forward-only coordinate optimization in the principal subspace of latent using the covariance matrix adaptation evolution strategy (CMA-ES). By updating a compact low-dimensional vector, TED not only enhances output confidence but also aligns the latent representation closer to the source latent distribution within the latent principal subspace. This is achieved without backpropagation, keeping the model parameters frozen, and enabling efficient, forgetting-free adaptation with minimal memory and computational overhead. Experiments on image classification and keyword spotting tasks across the ImageNet and Google Speech Commands series datasets demonstrate that TED achieves state-of-the-art performance while *reducing computational complexity by up to 63 times*, offering a practical and scalable solution for real-world edge applications. Furthermore, we successfully *deployed TED on the ZYNQ-7020 platform*, demonstrating its feasibility and effectiveness for resource-constrained edge devices in real-world deployments.

## 1 Introduction

The heterogeneity of data in real-world applications poses a significant challenge for modern machine learning systems. During deployment, the data encountered (*a.k.a.* target domain) often deviates from the training data (*a.k.a.* source domain), resulting in out-of-distribution (OOD) data (Recht et al., 2019; Hendrycks & Dietterich, 2019; Hendrycks et al., 2021). This distribution shift undermines the assumption of identical training and test distributions, causing models to struggle in generalizing effectively. OOD scenarios are particularly common in dynamic environments, where deployment conditions, sensor noise, and user behaviors vary significantly. Test-time adaptation (TTA) (Sun et al., 2020; Darestani et al., 2022; Liang et al., 2025) has emerged as a promising solution, allowing models to adapt dynamically to OOD data during inference, which is critical for ensuring robust and reliable AI systems in real-world settings.

The significance of TTA is heightened in edge computing, where AI models operate on resource-constrained devices such as FPGAs (Eldafrawy et al., 2020), ASICs (Yang et al., 2025), embedded platforms (Jeong et al., 2022), mobile devices (Li et al., 2024), and robots (Sodhani et al., 2021). While edge devices provide reduced latency, enhanced privacy, and real-time processing, their limited memory, computational power, and energy impose additional challenges for maintaining consistent OOD performance. Thus, developing TTA methods optimized for edge devices is essential, balancing adaptation efficacy with resource efficiency to enable robust and adaptive AI systems in diverse and dynamic deployment scenarios.

Many TTA approaches rely on gradient-based optimization to adjust model parameters during inference. For instance, pseudo-labeling (Liang et al., 2020) iteratively updates parameters based

on confident predictions, but its dependence on initial prediction quality can lead to performance degradation under severe distribution shifts. Other methods, such as TENT (Wang et al., 2021) and EATA (Niu et al., 2022), minimize self-supervised losses or impose constraints to stabilize adaptation, while MEMO (Zhang et al., 2022) enforces consistency across augmented samples. To further improve efficiency, recent works (Hong et al., 2023; Song et al., 2023; Lee et al., 2024; Ma et al., 2025) have proposed strategies to reduce computational overhead. Although effective in certain settings, these gradient-based approaches are unsuitable for resource-constrained edge devices due to their reliance on backpropagation, intermediate activation storage, and high computational overhead. Additionally, methods like MEMO, which adapt the entire model, are prone to catastrophic forgetting (Chen et al., 2025).

Gradient-free TTA methods have emerged as a promising alternative, leveraging lightweight updates to circumvent the limitations of gradient-based approaches. Many of these methods focus on adjusting batch normalization (BN) parameters (Schneider et al., 2020; Lim et al., 2023) or modifying output probabilities using batch-derived statistics (Boudiaf et al., 2022), but their learning capacity is limited. Moreover, in real-world edge device applications, such as image classification or keyword spotting, models typically encounter **independent single test sample rather than mini-batches of data**, rendering these batch-dependent methods impractical. Methods like T3A (Iwasawa & Matsuo, 2021) avoid batch dependency by adjusting the classifier directly; however, they perform poorly when adapting to individual test samples. While a recent prompt-based method FOA (Niu et al., 2024) eliminates backpropagation in a forward-only manner, we argue that FOA may be 1) suboptimal for independent single-instance adaptation due to potential reliance on batch statistics and 2) incompatible with wider prompt-free architectures (e.g., RNNs). These limitations highlight the need for robust gradient-free TTA methods that can handle single-instance scenarios and diverse architectures, underscoring the importance of further innovation in this area.

To this end, we introduce TED, a single-instance TTA method for edge devices that performs forward-only optimization in the latent principal subspace. Instead of tuning hundreds of parameters or entire models, TED updates only a low-dimensional vector. Unlike FOA's prompt updates, TED operates in an architecture-agnostic latent space, offering broader applicability and plug-and-play deployment. This yields high efficiency, strong adaptation, reduced forgetting, and reliable scaling on resource-limited hardware. Specifically, we pre-load the latent PC basis, through the SVD of the source latent representations. When an OOD test sample is fed into the model's encoder, it produces its corresponding latent. We then employ the Covariance Matrix Adaptation Evolution Strategy (CMA-ES) (Hansen, 2016) to update one compact vector, obtaining the adapted latent. During this process, entropy minimization is utilized to enhance the confidence of the final prediction, and the latent is modified closer to the source latent distribution within the latent principal subspace. Finally, the decoder generates the prediction based on the adapted latent.

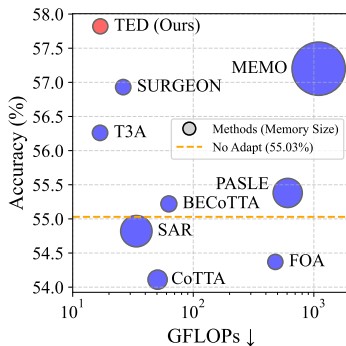

Figure 1: Accuracy, computation, and memory comparison of various TTA methods under a single-instance setting on ImageNet-C with ViT-Base.

Our main contributions are as follows: 1) **A Minimalist Latent Adaptation Paradigm:** We propose a novel TTA framework that shifts the adaptation focus from high-dimensional model weights to a low-dimensional latent space. We demonstrate that adapting only a handful of latent scalars is sufficient to correct distribution shifts. This paradigm decouples the adaptation complexity from the backbone size, effectively mitigating catastrophic forgetting while maintaining high robustness (see Figure 1). 2) **Forward-Only TTA Formulation:** By reducing the adaptation search space to a tiny latent vector, we are able to formulate TTA as a gradient-free optimization problem. This enables a forward-only update mechanism (implemented via CMA-ES) that eliminates the need for backpropagation and large activation buffers, making deep test-time adaptation feasible on strict edge devices for the first time. 3) **Efficiency and Universality:** We validate TED across five datasets involving significant real-world noise and distribution shifts. Unlike methods restricted to specific layers (e.g., Batch Norm), our approach is architecture-agnostic. Extensive experiments show that TED reduces computational complexity by up to $63\times$ and memory usage by $11\times$ compared to standard baselines, achieving state-of-the-art performance in single-instance TTA settings.

## 2 RELATED WORK

**Single-Instance TTA.** Single-instance TTA methods aim to adapt models to distribution shifts when only a single test sample is available, a scenario where the absence of batch data poses significant challenges for computing reliable statistics, especially for BN layers. To address this limitation, approaches like SITA (Khurana et al., 2021), DUA Mirza et al. (2022) MEMO (Zhang et al., 2022), and SPACE (Luo et al., 2025) generate a pseudo-batch by applying diverse augmentations to the single test sample. SITA adapts the parameters of BN layers based on the augmented batch, while MEMO fine-tunes the entire model to enforce consistency among the augmented samples. SPACE refines the model's encoder by aligning latent representations across the batch. However, from a hardware perspective, performing multiple augmentations introduces substantial challenges in terms of computational resources, overhead, and latency. Additionally, MEMO and SPACE rely on gradient-based optimization, making it unsuitable for deployment on resource-constrained edge devices.

**Gradient-Free TTA.** Gradient-free TTA methods address the computational and memory limitations of backpropagation, making them suitable for resource-constrained environments. Early studies in this area primarily focused on adapting BN statistics by recalculating the mean and variance from test data (Schneider et al., 2020). While effective in certain scenarios, these methods rely on the presence of multiple test samples, which limits their applicability in single-instance settings. To overcome this limitation, subsequent works have introduced techniques tailored for single-sample adaptation, such as SITA (Khurana et al., 2021), mix-up training (Hu et al., 2021), and instance-specific BN adjustments (Gong et al., 2022). In addition to BN adaptation, alternative strategies have been proposed, including prototype-based classifier adjustments (Iwasawa & Matsuo, 2021) and logit-level corrections (Boudiaf et al., 2022). Despite their computational efficiency, gradient-free TTA methods often suffer from limited learning capacity as they do not update the core model parameters, resulting in suboptimal performance under severe distribution shifts. These challenges underscore the need for more advanced gradient-free TTA approaches that can achieve a better balance between computational efficiency and adaptation effectiveness.

**Latent Representation Modification for TTA.** The modification of latent representations has been widely explored in image compression (Djelouah & Schroers, 2019; Shen et al., 2023) and generative modeling (Shen et al., 2020; Vahdat et al., 2021), where latent space manipulation has proven effective for improving task performance and flexibility. Existing TTA methods, however, rarely focus on directly modifying latent representations. A notable exception is (Chen et al., 2025), which introduces latent refinement for TTA in medical image segmentation using a latent conditional random field (CRF) loss. While effective, this approach relies on backpropagation, making it computationally expensive and unsuitable for edge computing. Moreover, its task-specific design for medical image segmentation and significant resource overhead limit its generalizability and practicality. These limitations highlight the need for efficient, lightweight, and generalizable TTA methods that modify latent representations without excessive computational costs.

## 3 METHODOLOGY

### 3.1 CHALLENGES AND MOTIVATION

**Challenges.** TTA aims to enable models to adapt dynamically to distribution shifts between source and target domain data during inference. Existing TTA methods face critical limitations on resource-constrained edge devices. Gradient-based methods (Wang et al., 2021) require backpropagation and substantial memory for storing the intermediate activations. Batch-dependent methods (Zhao et al., 2023) needs multiple samples, but edge applications often process single instances. Parameter-heavy approaches (Zhang et al., 2022) risk catastrophic forgetting and exceed memory constrains.

**Motivation.** We observe that distribution shifts primarily manifest as coordinate distortions when the test sample is projected into the source domain's semantic space. Instead of adapting model parameters, we propose to correct the latent representation of test sample by adjusting its coordinate within the source domain's principal subspace, which is spanned by the top-$k$ principal components (PC) of source latent feature.

Our approach offers three key advantages: 1) **Efficiency**: Only $k$ parameters need optimization ($k \ll D$, $D$ is the dimension of latent features). 2) **Preservation**: Source domain knowledge re-

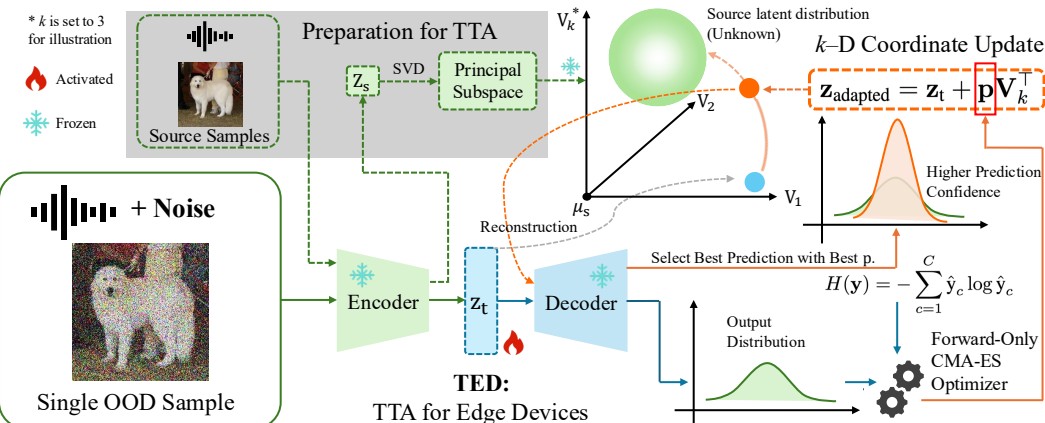

Figure 2: **An overview of our proposed TTA method for edge devices (TED)**. Source samples are used to compute the latent PC basis $\mathbf{V}_k$ during the preparation phase. For a single OOD sample, its latent is updated within the source latent principal subspace by encouraging higher prediction confidence and aligning it closer to the source latent distribution. This is achieved using a forward-only CMA-ES optimizer, enabling efficient and hardware-friendly TTA.

mains intact, avoiding catastrophic forgetting. 3) **Hardware-friendly**: Design for single-instance scenarios and no backpropagation required, enabling deployment on edge devices. Figure 2 illustrates the overall process of the proposed TED.

## 3.2 TED: EFFICIENT SINGLE-INSTANCE FORWARD-ONLY TEST-TIME ADAPTATION

**Definition 1.** (*Model and Latent Feature Representation*) Consider a model $f = \text{Dec} \circ \text{Enc}$, where the encoder $\text{Enc} : \mathcal{X} \rightarrow \mathcal{Z}$ may be instantiated by various architectures (e.g., Transformer, CNN, or LSTM), and the decoder $\text{Dec} : \mathcal{Z} \rightarrow \mathcal{Y}$ is a fully connected layer (or a variant thereof). For any input $\mathbf{x} \in \mathcal{X}$, the latent feature representation is defined as $\mathbf{z} := \text{Enc}(\mathbf{x})$, i.e., the input to the decoder; the model output is $\hat{\mathbf{y}} := \text{Dec}(\mathbf{z})$.

**Source Principal Subspace Construction.** Our framework is built upon representing latent within a subspace defined by the source domain's statistics. Given $N$ source latent features $\mathbb{Z}_s = \{\mathbf{z}_{s,i}\}_{i=1}^{N}$, where $\mathbf{z}_{s,i} \in \mathbb{R}^D$. First, we compute the source feature mean $\boldsymbol{\mu}_s$ and the centered latent $\mathbf{Z}_{s,\text{centered}}$.

We then perform truncated SVD to extract the $k$-dimensional principal subspace of source latent:

$$\mathbf{Z}_{s,\text{centered}} \approx \mathbf{U}_k \boldsymbol{\Sigma}_k \mathbf{V}_k^\top. \tag{1}$$

Here, $\mathbf{V}_k \in \mathbb{R}^{D \times k}$ is a matrix whose $k$ columns $\mathbf{v}_1, \mathbf{v}_2, \ldots, \mathbf{v}_k$ form an orthonormal basis for the $k$-dimensional principal subspace, which contains the $k$ principal directions capturing dominant source variation. The matrices $\mathbf{U}_k \in \mathbb{R}^{N \times k}$ and $\boldsymbol{\Sigma}_k \in \mathbb{R}^{k \times k}$ contain the corresponding left singular vectors and singular values respectively. The source-trained decoder is inherently optimized to perform well for source latent which is well-represented in the principal subspace. This low-rank basis $\mathbf{V}_k$ thus constitutes the "language" of semantic variation that the model understands.

**Coordinate Correction Framework and Theoretical Analysis.** Given the latent PC basis $\mathbf{V}_k$, any target latent $\mathbf{z}_t \in \mathbb{R}^D$ can be *approximated* as a deviation from the source mean, reconstructed from its projection onto the low-rank subspace:

$$\mathbf{z}_t \approx \boldsymbol{\mu}_s + \mathbf{p}_t \mathbf{V}_k^\top, \tag{2}$$

where $\mathbf{p}_t = (\mathbf{z}_t - \boldsymbol{\mu}_s) \mathbf{V}_k$ represents the $k$-dimensional vector of projection coefficients, or "coordinates", within the source latent principal subspace. Our proposed core latent adaptation strategy is formulated through the following update rule:

$$\mathbf{z}_{\text{adapted}} = \mathbf{z}_t + \mathbf{p} \mathbf{V}_k^\top, \tag{3}$$

where $\mathbf{p} \in \mathbb{R}^k$ is optimized during test time. By leveraging this framework, the complex problem of test-time domain adaptation is reformulated into a well-posed coordinate correction task within

---

**Algorithm 1** TED via Forward-Only Optimization in Latent Principal Subspace

---

1: **Input:** Test sample $\mathbf{x}$, encoder $\mathrm{Enc}$, decoder $\mathrm{Dec}$, latent PC basis $\mathbf{V}_k$, No. of iteration $n$.
2: **Output:** Prediction $\hat{\mathbf{y}}^*$.
3: **Step 1: Generate latent representation.**
4: Obtain latent $\mathbf{z}_t$ by passing the test sample $\mathbf{x}$ through the encoder: $\mathbf{z}_t = \mathrm{Enc}(\mathbf{x})$.
5: **Step 2: Optimize latent adaptation.**
6: Initialize CMA-ES optimizer.
7: **for** $t = 1$ **to** $n$ **do**
8:    **Sampling:** Generate $\lambda$ candidate solutions.
9:    **Evaluation:** For each candidate $\mathbf{p}_i^{(t)}$, compute the adapted latent by Equation 3,
10:    Obtain the output: $\hat{\mathbf{y}} = \mathrm{Dec}(\mathbf{z}_{\mathrm{adapted}})$ and compute the fitness using Equation 8.
11:    **Update:** Adapt CMA-ES internal parameters based on the top-performing candidates.
12: **end for**
13: **Step 3: Select final prediction.**
14: Choose the $\mathbf{p}^*$ with the smallest fitness value and corresponding output $\hat{\mathbf{y}}^*$.
15: **Return:** Final prediction $\hat{\mathbf{y}}^*$.

---

a canonical subspace defined by the source domain. This approach is computationally efficient and particularly suited for addressing distribution shifts.

Our core argument is that the adaptation rule in Equation 3 is mathematically equivalent to correcting the coordinates of the target latent within the unified source space. A source-trained model primarily interprets latent by its deviation from the source mean $\boldsymbol{\mu}_s$. Therefore, we analyze the deviation vector of the adapted latent:

$$\mathbf{z}_{\mathrm{adapted}} - \boldsymbol{\mu}_s = (\mathbf{z}_t - \boldsymbol{\mu}_s) + \mathbf{p}\mathbf{V}_k^\top. \tag{4}$$

This equation reveals that our method corrects the deviation vector of the target latent $(\mathbf{z}_t - \boldsymbol{\mu}_s)$ by adding a correction term $\mathbf{p}\mathbf{V}_k^\top$ that lies within the source latent PC space. To observe the effect in the coordinate space, we project Equation 4 onto the PC basis $\mathbf{V}_k$ by right-multiplying by $\mathbf{V}_k$:

$$(\mathbf{z}_{\mathrm{adapted}} - \boldsymbol{\mu}_s)\mathbf{V}_k = (\mathbf{z}_t - \boldsymbol{\mu}_s)\mathbf{V}_k + (\mathbf{p}\mathbf{V}_k^\top)\mathbf{V}_k. \tag{5}$$

Here, we define the coordinates as follows:

$$\mathbf{p}_{\mathrm{adapted}} = (\mathbf{z}_{\mathrm{adapted}} - \boldsymbol{\mu}_s)\mathbf{V}_k, \quad \mathbf{p}_{t \to s} = (\mathbf{z}_t - \boldsymbol{\mu}_s)\mathbf{V}_k, \tag{6}$$

where $\mathbf{p}_{\mathrm{adapted}}$ represents the coordinates of the adapted latent, and $\mathbf{p}_{t \to s}$ denotes the coordinates of the original target latent as observed in the source space. Since $\mathbf{V}_k^\top \mathbf{V}_k = \boldsymbol{I}$, the equation simplifies to the following coordinate correction formula:

$$\mathbf{p}_{\mathrm{adapted}} = \mathbf{p}_{t \to s} + \mathbf{p}. \tag{7}$$

This result demonstrates that our update rule reduces to a simple linear correction of the target latent's coordinates within the latent principal subspace. The following optimization of $\mathbf{p}$ drives this correction, effectively addressing the distribution shift through a unified mechanism.

**Forward-Only Optimization.** In the absence of ground-truth labels for the test sample, we adopt Shannon entropy (Shannon, 1948) minimization as the objective for TTA, a commonly used approach to encourage more confident model predictions (Grandvalet & Bengio, 2004; Wang et al., 2021; Zhang et al., 2022; Chen et al., 2025). The Shannon entropy is defined as:

$$H(\mathbf{y}) = -\sum_{c=1}^{C} \hat{\mathbf{y}}_c \log \hat{\mathbf{y}}_c, \tag{8}$$

where $\hat{\mathbf{y}}_c$ is the predicted probability for class $c$, and $C$ is the total number of classes. The optimization aims to minimize $H(\mathbf{y})$ with respect to $\mathbf{p}$, which drives the OOD latent feature closer to the source domain in $\mathbf{V}_k$ (see **Appendix A**).

To optimize $\mathbf{p}$ in a gradient-free manner, we employ CMA-ES, a powerful optimization algorithm designed for non-differentiable, multi-dimensional problems (see **Appendix B**). To ensure consistency in the optimization process for each test sample while accounting for computation cost, we

introduce a hyperparameter $n$ to explicitly control the number of optimization iterations. CMA-ES iteratively samples candidate solutions for $\mathbf{p}$, evaluates their fitness using the defined objective $H(\mathbf{y})$, and updates the search distribution. At the end, the prediction output corresponding to the $\mathbf{p}^*$ with the smallest fitness value is selected. The overall method is presented in Algorithm 1.

Overall, our method bridges the gap between algorithmic performance and hardware deployment, providing a robust and efficient framework for TTA on edge devices. More discussion is presented in **Appendix A**. The code will be available upon the acceptance.

## 4 EXPERIMENTS

### 4.1 EXPERIMENTS SETUP

**Tasks, Datasets, and Models.** We evaluate our methods on two types of tasks: image classification (IC) and keyword spotting (KWS). For the IC task, we conduct experiments on four OOD generalization benchmarks: ImageNet-C (Hendrycks & Dietterich, 2019), ImageNet-V2 (Recht et al., 2019), ImageNet-R (Hendrycks et al., 2021), and ImageNet-Sketch (Wang et al., 2019). We use ViT-Base (Dosovitskiy et al., 2021), trained on ImageNet-1k (Russakovsky et al., 2015), as pretrained source model across all four datasets. For the KWS task, we simulate real-world scenarios by mixing the Google Speech Commands dataset (Warden, 2018) with five types of real-world background noise from the ESC50 dataset (Piczak, 2015) at varying signal-to-noise ratios (SNR), referred as GSC-C. The source model used is a pretrained LSTM (Yang et al., 2025), trained on the clean GSC dataset. The evaluation metric is the classification **accuracy** $(\%, \uparrow)$ on OOD test-time samples.

**Baselines.** We evaluate our method against two types of baselines: gradient-free and gradient-based methods, as well as a simple baseline, No Adapt, which performs no TTA. Gradient-free methods include T3A (Iwasawa & Matsuo, 2021), which adapts a prototype-based classifier to handle OOD samples, and FOA (Niu et al., 2024), which optimizes additional prompts without gradient updates for efficient adaptation. Gradient-based methods include CoTTA (Wang et al., 2022), which employs continual adaptation to enhance consistency across augmented samples, MEMO (Zhang et al., 2022), which leverages entropy minimization for confident predictions, SAR (Niu et al., 2023), which stabilizes TTA through active sample selection and a sharpness-aware optimizer, PASLE (Hu et al., 2025), which adapts progressively by assigning one-hot labels to confident samples and candidate sets to uncertain ones, BECoTTA (Lee et al., 2024), which utilizes input-dependent mixture-of-experts for parameter-efficient TTA, and SURGEON (Ma et al., 2025), which reduces memory cost via dynamic activation sparsity during backpropagation. These baselines encompass diverse strategies, ensuring a comprehensive comparison. All baselines in our experiments are reproduced using the official implementations and the hyper-parameters recommended in their original papers or public code repositories, except the batch size which is set to 1.

**Implementation Details.** For the IC and KWS tasks, we set $k$ to 16 and 2, respectively, and compute the source PC basis $\mathbf{V}_k$, which remains fixed throughout the optimization process. The population size $\lambda$ for CMA-ES initialization is set to $(4 + 3 \times \log k)$, following the default configuration of Hansen (2016). The number of optimization iterations $n$ is set to 8 for IC and 2 for KWS. To ensure a fair comparison in our single-instance setting, the batch size for all baselines is fixed at 1, and the model is reset after processing each test sample to maintain independence.

Additional details on the experimental setup and extended experiments can be found in **Appendix C**.

### 4.2 MAIN RESULTS AND ANALYSES

In this section, we evaluate our proposed TED method on two tasks: IC and KWS, comparing it against state-of-the-art TTA methods. The primary focus is to assess the effectiveness of our method in handling distribution shifts, while maintaining efficiency and stability during TTA. The results highlight the superior performance of our approach across diverse datasets and tasks.

**Performance Comparison on Image Classification Task.** Table 1 summarizes the performance of various methods on ImageNet-C with the ViT-Base model under diverse distribution shifts. We discuss the results in four key aspects: 1) **Superior Performance:** Our method, TED, achieves the highest average accuracy of 57.82%, surpassing all baselines, which highlights its robustness and

Table 1: Performance comparison on ImageNet-C with ViT-Base model regarding **Accuracy** (%). **GF** stands for gradient-free. The **bold** number indicates the best result.

| Method | GF | Noise | | | Blur | | | | Weather | | | | Digital | | | Average |
| | | Gauss. | Shot | Impl. | Defoc. | Glass | Motion | Zoom | Snow | Frost | Fog | Brit. | Contr. | Elas. | Pix. | JPEG | Acc. |
|---|---|---|---|---|---|---|---|---|---|---|---|---|---|---|---|---|---|
| No Adapt | ✓ | 55.34 | 56.23 | 56.01 | 46.48 | 34.78 | 52.87 | 44.20 | 62.39 | 62.66 | 65.56 | 77.70 | 32.04 | 45.73 | 66.72 | 66.67 | 55.03 |
| FOA (ICML'24) | ✓ | 53.87 | 54.16 | 54.00 | 46.17 | 33.45 | 52.56 | 43.69 | 61.82 | 62.30 | 66.17 | 77.73 | 30.60 | 46.14 | 66.18 | 66.77 | 54.37 |
| T3A (NeurIPS'21) | ✓ | 54.69 | 55.95 | 55.61 | 47.41 | 36.77 | 53.91 | 46.44 | 63.85 | 60.42 | 68.12 | 78.11 | 37.79 | 49.54 | 67.24 | 68.04 | 56.26 |
| CoTTA (CVPR'22) | ✗ | 54.61 | 55.66 | 55.37 | 45.28 | 34.35 | 52.69 | 44.11 | 62.38 | 62.62 | 58.33 | 77.71 | 29.58 | 45.65 | 66.68 | 66.66 | 54.11 |
| SAR (ICLR'23) | ✗ | 55.25 | 56.08 | 55.89 | 46.22 | 34.41 | 52.28 | 43.82 | 62.09 | 62.69 | 65.56 | 77.53 | 32.03 | 45.47 | 66.37 | 66.55 | 54.82 |
| PASLE (ICLR'25) | ✗ | 56.72 | 56.24 | 56.21 | 47.53 | 35.32 | 53.02 | 44.03 | 62.43 | 62.81 | 65.84 | 78.62 | 31.23 | 46.65 | 66.76 | 67.24 | 55.38 |
| BECoTTA (ICML' 2024) | ✗ | 55.67 | 56.45 | 56.29 | 46.21 | 33.68 | 52.66 | 43.67 | 62.20 | 63.37 | **68.25** | 77.58 | 33.74 | 45.09 | 66.70 | 66.78 | 55.22 |
| SURGEON (CVPR' 2025) | ✗ | 58.70 | 59.22 | 59.23 | 48.82 | 35.29 | 55.06 | 45.87 | 64.83 | 65.94 | 61.76 | 79.56 | 34.46 | 46.90 | 69.02 | 69.36 | 56.93 |
| MEMO (NeurIPS'22) | ✗ | 55.90 | 54.20 | 56.30 | 45.79 | **39.34** | 53.02 | 45.13 | 42.56 | 47.82 | 65.31 | 80.01 | **69.63** | **49.21** | 69.51 | **71.33** | 56.34 |
| TED (ours) | ✓ | **58.77** | **59.66** | **59.50** | **49.30** | 36.08 | **55.35** | **46.34** | **65.21** | **66.40** | 67.66 | **80.21** | 35.96 | 47.61 | **69.55** | 69.68 | **57.82** |

adaptability. TED consistently outperforms in and most individual domains, further demonstrating its ability to adapt effectively without requiring gradient updates. 2) **Setting Challenge:** Many methods, including FOA, CoTTA and SAR, fail to achieve meaningful improvements in single-instance adaptation scenarios, reducing their applicability in real-world settings where efficient and stable TTA is needed. FOA's activation-shifting module requires batch data to function reliably. Moreover, the "single-sample" variant in FOA actually relies on a continuous test-time stream, which contradicts our assumption that test instances arrive independently and must be handled in isolation, which better matches real deployments. CoTTA's EMA teacher is updated from single, noisy pseudo-labels and thus cannot supply reliably denoised targets, and augmentation-averaged labels are often disabled or too sparse to stabilize the update. SAR suffers from the combination of batch size = 1 and online label imbalance yields too few reliable samples for updates. 3) **Vs. T3A:** T3A relies on a history-dependent support set that is incrementally updated using previous test samples. As a result, its predictions are sequence-dependent and cannot be made independent across test instances. If per-instance independence is enforced (e.g., by resetting the support set for each input), each adjustment benefits only from the initialization and offers limited effective adaptation. 4). **Vs. PASLE:** PASLE helps by using selective labels—one-hot for confident cases and small candidate sets for uncertain ones—mitigating outright mislabeling. However, its strengths that rely on progressive thresholding, buffer-based reuse, and stable margin statistics are underutilized with batch size = 1, leading to limited but consistent gains, which is line with its report on batch size sensitivity in the paper. 5) **Vs. MEMO:** MEMO face challenges due to instability and catastrophic forgetting. Although MEMO achieves a competitive average accuracy, it exhibits significant inconsistencies across domains, such as Snow (42.56%) and Fog (47.82%). Overall, TED demonstrates state-of-the-art performance, superior robustness, and strong adaptability, making it highly effective for tackling diverse distribution shifts in real-world applications.

Beyond ImageNet-C, our method achieves superior performance on the ImageNet-V2, -R, and -Sketch datasets, as shown in Table 2, achieving the highest average accuracy of 63.72% and consistently outperforming all baselines. These results underscore TED's exceptional ability to adapt to distribution shifts across diverse data, further validating its robustness and strong generalizability.

**Performance Comparison on Keyword Spotting Task.** CoTTA and MEMO require standard image augmentation to perform TTA. However, their methods lack well-defined transformations tailored for speech data, making them unsuitable for a fair comparison in this task. Similarly, FOA, as a prompt-based method, is incompatible with LSTM ar-

Table 2: Performance comparison on ImageNet-V2/R/Sketch with ViT-Base regarding **Accuracy** (%). **GF** stands for gradient-free. The **bold** number indicates the best result.

| Method | GF | Accuracy (%) | | | |
| | | V2 | R | Sketch | Avg. |
|---|---|---|---|---|---|
| No Adapt | ✓ | 75.49 | 59.49 | 44.89 | 59.96 |
| FOA (ICML'24) | ✓ | 75.25 | 59.96 | 44.95 | 60.05 |
| T3A (NeurIPS'21) | ✓ | 75.61 | 57.98 | 48.44 | 60.68 |
| CoTTA (CVPR'22) | ✗ | 75.50 | 59.20 | 44.77 | 59.82 |
| SAR (ICLR'23) | ✗ | 75.33 | 59.39 | 44.82 | 59.85 |
| PASLE (ICLR'25) | ✗ | 75.66 | 61.73 | 45.72 | 61.04 |
| MEMO (NeurIPS'22) | ✗ | 76.08 | 62.85 | 46.08 | 61.67 |
| TED(ours) | ✓ | **78.15** | **65.29** | **47.73** | **63.72** |

chitectures, which are commonly used in KWS. Therefore, we compare our proposed method, TED, with T3A and SAR on the GSC-C dataset under SNR of -10/-15/-20 dB, as shown in Table 3. The results demonstrate that TED significantly outperforms other baselines, with performance improvements becoming more pronounced as the SNR decreases. We attribute this to the fact that under higher noise levels, the same principal subspace ($\mathbf{V}_k$) provides relatively more informative guidance from the source domain, enabling TED to perform more effective adaptation. Furthermore, we find that T3A's performance on KWS is comparable to No Adapt. We attribute this to the small label space (12 classes) and the single-instance setting, which yield too few confident per-class supports to update the prototypes; consequently, the pseudo-prototypes remain close to the initial classifier weights, the adjusted logits mirror the original linear head, and accuracy is unchanged. Because

Table 3: Performance comparison on GSC-C with LSTM model regarding **Accuracy** (%). **GF** stands for gradient-free. The **bold** number indicates the best result.

| SNR | Method | GF | Animals | | Natural | | Human | | Domestic | | Urban | | Average |
| | | | dog | cat | pouringwater | thunderstorm | cryingbaby | laughing | washingmachine | vacuumcleaner | carhorn | fireworks | Acc. |
|---|---|---|---|---|---|---|---|---|---|---|---|---|---|
| -10 dB | No Adapt | ✓ | 62.67 | 61.17 | 54.55 | 66.23 | 58.74 | 58.59 | 52.88 | 50.43 | 56.62 | 61.54 | 58.34 |
| | T3A (NeurIPS'21) | ✓ | 62.67 | 61.17 | 54.55 | 66.23 | 58.74 | 58.59 | 52.88 | 50.43 | 56.62 | 61.54 | 58.34 |
| | SAR (ICLR'23) | ✗ | 61.33 | 59.85 | 52.79 | 63.92 | 55.80 | 56.95 | 49.53 | 47.31 | 55.06 | 60.21 | 56.28 |
| | PASLE (ICLR'25) | ✗ | 62.87 | 62.06 | 54.78 | 66.75 | 59.42 | 59.63 | 57.69 | 53.38 | 58.35 | 62.73 | 59.77 |
| | TED (ours) | ✓ | **64.25** | **63.58** | **59.73** | **66.47** | **61.99** | **61.94** | **59.46** | **56.98** | **59.32** | **64.90** | **61.86** |
| -15 dB | No Adapt | ✓ | 57.08 | 53.63 | 49.35 | 61.45 | 53.08 | 53.03 | 46.81 | 47.49 | 51.76 | 55.19 | 52.89 |
| | T3A (NeurIPS'21) | ✓ | 57.08 | 53.63 | 49.35 | 61.45 | 53.08 | 53.03 | 46.81 | 47.49 | 51.76 | 55.19 | 52.89 |
| | SAR (ICLR'23) | ✗ | 55.33 | 52.36 | 47.49 | 59.35 | 51.01 | 50.85 | 44.13 | 44.37 | 50.82 | 54.13 | 50.98 |
| | PASLE (ICLR'25) | ✗ | 58.12 | 53.77 | 52.49 | 61.93 | 55.64 | 56.72 | 48.34 | 49.23 | 52.67 | 59.32 | 54.82 |
| | TED (ours) | ✓ | **60.84** | **57.99** | **57.71** | **62.28** | **58.04** | **58.98** | **57.83** | **55.38** | **56.00** | **60.41** | **58.55** |
| -20 dB | No Adapt | ✓ | 52.75 | 48.50 | 46.21 | 58.08 | 51.12 | 48.53 | 45.55 | 46.05 | 48.81 | 50.94 | 49.65 |
| | T3A (NeurIPS'21) | ✓ | 52.75 | 48.50 | 46.21 | 58.08 | 51.12 | 48.53 | 45.55 | 46.05 | 48.81 | 50.94 | 49.65 |
| | SAR (ICLR'23) | ✗ | 51.28 | 46.95 | 44.28 | 55.51 | 47.82 | 46.68 | 41.45 | 43.41 | 48.23 | 50.10 | 47.57 |
| | PASLE (ICLR'25) | ✗ | 53.76 | 51.77 | 48.84 | 59.03 | 53.31 | 50.38 | 46.21 | 50.21 | 52.17 | 53.56 | 51.92 |
| | TED (ours) | ✓ | **59.07** | **54.71** | **57.20** | **59.35** | **56.94** | **57.94** | **58.22** | **55.54** | **54.50** | **58.07** | **57.15** |

the LSTM backbone lacks normalization layers, SAR—which adapts only the affine parameters of group/layer norms—cannot implement the two-step "Reliable Sample Filtering + Sharpness-Aware Minimization" procedure. As a result, adaptation collapses to plain entropy minimization and is often ineffective or unstable. In line with our findings in the IC task, PASLE provides small but consistent gains, which further exposing the limitations of existing methods in real-world settings and underscoring the practical significance of our approach.

## 4.3 Ablation Studies

**Analyses of Computational Efficiency.** As shown in Figure 1 and Table 4, TED demonstrates significant advantages in computational complexity compared to other methods. Specifically, TED achieves a GFLOPs value of 16.95, which is among the lowest across all methods, highlighting its high computational efficiency. T3A suffers from longer runtime despite having the lowest GFLOPs, due to its computation being concentrated in the final linear layer and support set updates, which are difficult to parallelize and fully utilize hardware resources. Additionally, its entropy filtering step, which involves calculating and filtering prediction entropy for each sample, introduces additional overhead when the support set is large. In terms of memory usage, TED requires only

Table 4: GFLOPs, memory usage and running time per sample comparison on ImageNet-C with ViT-Base. **GF** stands for gradient-free. The **bold** number indicates the best result.

| Method | GF | GFLOPs | Mem (MB) | Time (s) |
|---|---|---|---|---|
| FOA (ICML'24) | ✓ | 479.31 | 702 | 0.273 |
| T3A (NeurIPS'21) | ✓ | **16.86** | 718 | 0.124 |
| CoTTA (CVPR'22) | ✗ | 50.59 | 1130 | 0.703 |
| SAR (ICLR'23) | ✗ | 33.73 | 2996 | **0.037** |
| PALSE (ICLR'25) | ✗ | 607.13 | 2588 | 0.051 |
| MEMO (NeurIPS'22) | ✗ | 1096.14 | 8632 | 1.009 |
| PALSE (ICLR'25) | ✗ | 607.13 | 2588 | 0.051 |
| BECoTTA (ICML' 2024) | ✗ | 62.74 | 778 | 0.082 |
| SURGEON (CVPR' 2025) | ✗ | 26.24 | 716 | 0.071 |
| MEMO (NeurIPS'22) | ✗ | 1096.14 | 8632 | 1.009 |
| TED(ours) | ✓ | 16.95 | **696** | 0.042 |

696 MB, making it the most memory-efficient approach in the comparison. Moreover, TED achieves a short runtime per sample, at just 0.042 seconds, significantly outperforming other methods such as MEMO (1.009 s) and CoTTA (0.703 s). Gradient-based SAR achieve slightly shorter running time by only updating the affine parameters in normalization layers, thereby reducing the computational cost of parameter updates. However, as shown in Table 1, this strategy struggles in single-sample scenarios, where updating affine parameters alone may not be sufficient to achieve effective TTA.

**Effectiveness of TED strategy.** We analyze 1000 ImageNet-C (Gaussian Noise) samples, and measure the Cosine Similarity between the latent features of OOD samples and their corresponding clean source features Figure 3. TED-adapted features show higher similarity to the source compared to the corrupted features, indicating effective recovery of the original feature semantics.

**Effect on Diverse Networks.** To evaluate the generalizability of TED, we validate its effectiveness on diverse architectures, including ResNet-50 (He et al., 2016), EfficientNet-B0 (Tan & Le, 2019), and MobileNet-V4 (Qin et al., 2024). Since these backbones rely on Batch Normalization, we incorporate additional BN-specific baselines IABN (Gong et al., 2022) and TTN (Lim et al., 2023) for comparison in Table 5.

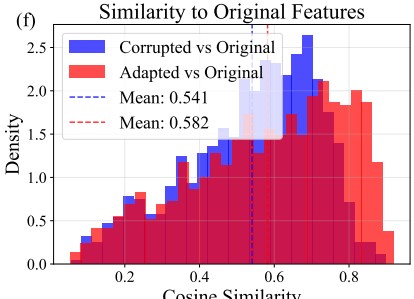

Figure 3: Visualization of latent feature alignment, regarding Cosine Similarity.

Table 5: Performance comparison on ImageNet-C with ResNet-50, EfficientNet-B0 and MobileNet-V4 regarding **Accuracy** (%). The **bold** number indicates the best result.

| Networks | Method | Noise | | | Blur | | | | Snow | Weather | | | | Digital | | | Average |
|---|---|---|---|---|---|---|---|---|---|---|---|---|---|---|---|---|---|
| | | Gauss. | Shot | Impl. | Defoc. | Glass | Motion | Zoom | | Frost | Fog | Brit. | Contr. | Elas. | Pix. | JPEG | Acc. |
| ResNet-50 | No Adapt | 4.47 | 4.74 | 4.06 | 8.11 | 5.96 | 9.59 | 14.74 | 6.92 | 14.47 | 12.32 | 45.80 | 0.72 | 11.08 | 18.19 | 32.54 | 12.91 |
| | IABN | **7.22** | 5.10 | **5.54** | 5.46 | **7.38** | 11.92 | 15.98 | 5.74 | 12.32 | 14.14 | 47.10 | **0.90** | 7.10 | 18.26 | 18.04 | 12.15 |
| | TTN | 1.73 | 5.13 | 4.90 | 4.17 | 6.69 | 10.16 | 14.14 | 6.13 | 14.52 | 14.42 | 49.20 | 0.22 | 12.26 | 22.03 | 33.07 | 13.25 |
| | SAR | 2.34 | 4.46 | 4.24 | 6.56 | 6.66 | 14.04 | 16.34 | 4.02 | 14.68 | 13.26 | 48.51 | 0.35 | 12.43 | 20.44 | 36.46 | 13.65 |
| | TED | 4.99 | **5.18** | 4.41 | **11.36** | 7.32 | **14.26** | **18.29** | **7.00** | **15.32** | **15.01** | **55.32** | 0.25 | **13.07** | **23.29** | **40.97** | **15.74** |
| EfficientNet-B0 | No Adapt | 15.07 | 18.47 | 14.66 | 21.48 | 8.66 | 21.73 | 24.24 | 30.95 | 27.77 | 28.80 | 67.25 | 21.96 | 17.62 | 46.73 | 50.51 | 27.73 |
| | IABN | 14.71 | 19.32 | 14.75 | 21.34 | **9.24** | 21.59 | 25.65 | 33.35 | 29.45 | 29.56 | 69.24 | 22.51 | 17.88 | 49.46 | 53.32 | 28.76 |
| | TTN | 15.32 | 12.34 | 15.32 | 23.44 | 6.32 | 23.23 | 26.36 | 32.02 | 28.66 | 32.74 | 69.35 | 24.78 | 18.50 | 52.64 | 54.44 | 29.03 |
| | SAR | 15.34 | 20.58 | 16.22 | 22.53 | 4.24 | 20.34 | 22.35 | 32.45 | 20.76 | 28.63 | 68.64 | 22.46 | 16.34 | 46.86 | 49.58 | 27.15 |
| | TED | **16.61** | **21.53** | **16.71** | **25.02** | 7.57 | **25.91** | **27.69** | **35.05** | **30.38** | **32.78** | **75.67** | **25.96** | **18.50** | **55.14** | **59.24** | **31.58** |
| MobileNet-V4 | No Adapt | 5.97 | **7.58** | 6.02 | 9.69 | **3.03** | 12.30 | 11.19 | 12.57 | 11.41 | 20.11 | 58.11 | **3.84** | 11.05 | 10.29 | 32.79 | 14.40 |
| | IABN | 5.32 | 7.50 | 6.23 | 9.46 | 3.02 | 12.34 | 11.68 | 12.34 | 20.68 | **13.46** | 66.48 | 2.02 | 11.36 | 9.68 | 36.46 | 15.20 |
| | TTN | **6.98** | 7.54 | **6.38** | 9.84 | 3.34 | 13.56 | **12.56** | 12.64 | 20.46 | 12.42 | 64.66 | 2.46 | 10.48 | 10.46 | 40.58 | 15.62 |
| | SAR | 4.98 | 7.32 | 4.54 | 10.20 | 3.02 | 14.54 | 11.08 | 12.78 | 22.34 | 11.60 | 60.34 | 2.66 | 10.46 | 11.42 | 39.64 | 15.13 |
| | TED | 5.47 | 7.55 | 5.43 | **10.89** | 2.99 | **14.57** | 12.28 | **13.17** | **23.08** | 12.70 | **68.52** | 2.78 | **11.77** | **11.77** | **41.43** | **16.29** |

Table 6: Performance comparison on ImageNet-C with ViT-Base model using different $k$ and $n$ regarding **Accuracy** (%). The **bold** number indicates the best result.

| k$\{k\}$n$\{n\}$ | k8n2 | k8n4 | k8n8 | k8n10 | k16n2 | k16n4 | k16n8 | k16n10 | k32n2 | k32n4 | k32n8 | k32n10 |
|---|---|---|---|---|---|---|---|---|---|---|---|---|
| Accuracy | 55.07 | 55.39 | 51.40 | 50.01 | 55.12 | 56.88 | **57.82** | 52.44 | 55.16 | 55.56 | 57.22 | 53.24 |

On ResNet-50, TED achieves consistent performance improvements across most domains, with a notable +2.83% increase in average accuracy. Furthermore, considering edge deployment, we examine the lightweight EfficientNet-B0 (5.29M parameters) and MobileNet-V4 (3.77M parameters). Despite the inherently lower robustness of these smaller networks, TED still improves the average accuracy. However, we observe minimal gains on specific extreme corruptions, such as Glass Blur on MobileNet-V4 and Contrast on ResNet-50. In these cases, the baseline accuracy drops to near-random levels (e.g., 3.03% and 0.72%, respectively), indicating catastrophic feature degradation. Since TTA relies on exploiting residual semantic structures in the latent space, the total loss of meaningful signal in these regimes renders the adaptation ineffective. This suggests that under such severe information loss, improving the intrinsic robustness of the backbone is a prerequisite for successful adaptation.

**Effect of Hyperparameters $k$ and $n$.** As shown in Table 6, the choice of $k$ and $n$ presents a trade-off between subspace expressiveness and optimization stability, with $k = 16, n = 8$ achieving the optimal balance. Specifically, $k$ controls the subspace capacity: a small $k$ limits useful variation, while an excessively large $k$ introduces noisy directions and complicates the search space. Regarding $n$, it governs the optimization strength. While sufficient steps are needed for adaptation, an overly large $n$ causes the solver to overfit the unsupervised proxy rather than improving generalization, leading to performance degradation.

Table 7: Evaluation on edge device for KWS task (GSC-C under SNR of -10 dB) with LSTM model regarding **Accuracy** (%). The **bold** number indicates the best result.

| Method | Devices | Animals | | Natural | | Human | | Domestic | | Urban | | Average |
|---|---|---|---|---|---|---|---|---|---|---|---|---|
| | | dog | cat | pouringwater | thunderstorm | cryingbaby | laughing | washingmachine | vacuumcleaner | carhorn | fireworks | Acc. |
| No Adapt | RTX 3090 | 62.67 | 61.17 | 54.55 | 66.23 | 58.74 | 58.59 | 52.88 | 50.43 | 56.62 | 61.54 | 58.34 |
| TED | | **64.25** | **63.58** | **59.73** | **66.47** | **61.99** | **61.94** | **59.46** | **56.98** | **59.32** | **64.90** | **61.86** (+3.52) |
| No Adapt | ZYNQ 7020 | 57.03 | 56.29 | 50.03 | 60.74 | 51.52 | 52.42 | 50.75 | 52.88 | 54.74 | 59.19 | 54.56 |
| TED | | **58.06** | **57.70** | **53.04** | **61.04** | **53.63** | **54.31** | **52.39** | **54.93** | **58.18** | **59.62** | **56.29** (+1.73) |

**Demonstration on Edge Device ZYNQ 7020.** To validate the feasibility of our TTA framework on real-world edge devices, we deployed it on the ZYNQ 7020 platform, a widely utilized SoC that combines an ARM Cortex-A9 processor with FPGA-based programmable logic. We frame this experiment as a proof-of-concept for edge deployability rather than a conventional benchmark, given that the strict hardware constraints preclude the use of standard gradient-based TTA methods. While most existing approaches rely on computationally expensive backpropagation and large activation buffers, rendering them incompatible with such platforms, TED overcomes this barrier by updating only a low-dimensional latent vector via forward passes.

We evaluated our algorithm on the KWS task under an SNR of -10 dB. The reduced computational precision on ZYNQ 7020 (fixed-point 16-bit) contributes to the performance gap observed between

the edge device and the GPU (float-point 32-bit), as shown in Table 7. Nevertheless, our TTA method achieves notable performance improvements, with an average accuracy of 54.56% compared to 56.29% for the baseline. These results underscore the robustness and adaptability of our framework, even under the constraints of edge hardware, positioning it as a promising solution for real-world deployment in resource-constrained environments.

## 5 CONCLUSION

In this paper, we proposed a TTA framework that updates the latent representation of a single test sample within the principal subspace, achieving robust classification performance with high computational efficiency. By employing the forward-only CMA-ES optimizer, our method is particularly well-suited for edge devices. We validated our approach across five datasets with significant distribution shifts, demonstrating reductions in computational complexity and resource usage, while achieving state-of-the-art performance in single-instance TTA. Additionally, we incorporated quantization techniques to further enhance the hardware efficiency of our method. To validate its real-world applicability, we successfully deployed our method on the ZYNQ 7020 platform, showcasing its feasibility and effectiveness in practical edge computing scenarios.

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

## A  DISCUSSION

**Decomposition of Distribution Shift.** We interpret our method within a "Unified Centered Space", where the origin is the source mean $\boldsymbol{\mu}_\text{s}$ and the axes are defined by the latent PC basis $\mathbf{V}_k$. The distribution shift between the source and target domains can be decomposed into two distinct components: **Mean Shift**, representing the difference in data centroids, $\Delta\boldsymbol{\mu} = \boldsymbol{\mu}_\text{t} - \boldsymbol{\mu}_\text{s}$, and **Covariance Shift**, capturing changes in the data distribution's shape, orientation, and scale. The latter is reflected in the mismatch of PC and the differing distributions of projection coordinates. A target latent $\mathbf{z}_\text{t}$ poses a challenge for the source model as it is simultaneously influenced by both types of shifts.

**Mechanism of Coordinate Correction.** Because the update vector $\mathbf{p}$ is searched *only* inside the source principal subspace $\mathbf{V}_k$, every candidate latent is expressed in a coordinate system that the source-trained decoder inherently supports. The Shannon entropy objective $H$ then acts as a directional guide. Our rationale relies on standard modeling assumptions regarding deep classifiers: specifically, that softmax outputs are calibrated (Guo et al., 2017) and that class-conditional features follow a shared-covariance prototype geometry (Lee et al., 2018; Papyan et al., 2020). Under these conditions, minimizing entropy encourages $\mathbf{z}_\text{adapted}$ to move toward the high-density neighborhood of a source class prototype. In the Unified Centered Space, this movement re-centers the sample toward a source class mean and reduces the Mahalanobis residual, effectively mitigating covariance shift. This theoretical analysis is corroborated by our empirical results in Figure 3, which verify that the adapted latent statistics align closely with the clean source distribution. Consequently, the combination of "subspace restriction + entropy minimization" allows us to pull OOD features back to the source distribution **without any explicit distance regularizer**, achieving effective test-time adaptation with minimal overhead.

**Quantization of TED.** While the computation in our primary ZYNQ deployment is automatically truncated by the hardware's arithmetic units, we further explore the limits of TED's efficiency to demonstrate its potential for future dedicated hardware co-design. Specifically, we investigate whether the optimization process remains effective under extreme numerical constraints. We propose two exploratory variants

1. **Definition 2.** (*QTED-V1*) We quantize the optimization target $\mathbf{p}$ after each iteration into a 1-bit representation, where each element of $\mathbf{p}$ can assume only one of two possible values. From a hardware perspective, this approach reduces the optimization process to modifying the states of $k$ binary switches, which significantly lowers computational complexity and memory requirements. This streamlined representation of $\mathbf{p}$ minimizes the overhead associated with updates during TTA.

2. **Definition 3** (*QTED-V2*) Based on QTED-V1, to address the absence of high-precision floating-point support in certain hardware environments, we further simulate the CMA-ES process using fixed-point arithmetic. This quantization approach ensures compatibility with constrained hardware while preserving the effectiveness of the optimization process.

QTED-V1 and QTED-V2 demonstrate the adaptability of our algorithm to diverse hardware architectures, making it well-suited for resource-limited edge applications.

**Future Work.** Our current contribution focuses on an algorithmic design that is mindful of hardware constraints, yielding an efficient and deployable TTA solution. Moving forward, we will pursue algorithm–hardware co-design, exploring hardware-level optimizations alongside TTA-dedicated accelerator modules. These efforts aim to further reduce latency and memory footprint, enhance energy efficiency, and strengthen practical performance on deployed systems.

## B  COVARIANCE MATRIX ADAPTATION EVOLUTION STRATEGY

Considering the applicability of our TTA method on resource-constrained edge devices and the fact that our approach only requires updating a very small number of parameters, we adopt the Covariance Matrix Adaptation Evolution Strategy (CMA-ES) (Hansen, 2016) as our optimizer. CMA-ES is a gradient-free, population-based optimization algorithm that is particularly well-suited for non-differentiable, black-box problems and multi-dimensional search spaces.

Table 8: Five backbone models and their hidden size $D$ (first dimension of the latent PC basis $\mathbf{V}_k$).

| Model | ViT-Base | ResNet-50 | EfficientNet-B0 | MobileNet-V4 | LSTM |
|---|---|---|---|---|---|
| $D$ | 768 | 2048 | 1280 | 1280 | 32 |

The optimization process of CMA-ES in each iteration begins by initializing a multivariate Gaussian search distribution $\mathcal{N}(\mathbf{m}^{(t)}, \sigma^{(t)}\mathbf{\Sigma}^{(t)})$, where $\mathbf{m}^{(t)}$ is the current mean, $\sigma^{(t)}$ is the step size, and $\mathbf{\Sigma}^{(t)}$ is the covariance matrix at iteration $t$. During each iteration, a population of candidate solutions $\{\mathbf{p}_i^{(t)}\}_{i=1}^{\lambda}$ is sampled from this distribution according to the rule:

$$\mathbf{p}_i^{(t)} \sim \mathbf{m}^{(t)} + \sigma^{(t)}\mathcal{N}(\mathbf{0}, \mathbf{\Sigma}^{(t)}). \tag{9}$$

Each candidate solution $\mathbf{p}_i^{(t)}$ is then evaluated using the predefined fitness function, which in our case is the Shannon entropy $H(\mathbf{y})$ of the model's output. Based on the fitness values, the mean $\mathbf{m}^{(t)}$ is updated to reflect the top-performing candidates, and the covariance matrix $\mathbf{\Sigma}^{(t)}$ is adapted to better capture the structure of the search space.

CMA-ES is particularly suitable for our scenario as it avoids gradient computation entirely, reducing the computational overhead on devices that are unable to support backpropagation. Furthermore, its iterative sampling and distribution adaptation effectively explore the low-dimensional parameter space of $\mathbf{p}$, making it efficient even under strict resource constraints.

## C  MORE EXPERIMENTAL DETAILS

### C.1  MORE DETAILS ON DATASET

**ImageNet-C (Hendrycks & Dietterich, 2019).** ImageNet-C is a standardized benchmark for assessing the robustness of image classifiers to common distribution shifts. It applies 15 algorithmically generated, label-preserving corruptions to the 50,000 images in the ImageNet-1k validation set, each at five severity levels, yielding 75 corrupted test sets (3.75 million images). The corruptions span four categories: noise (Gaussian, shot, impulse), blur (defocus, glass, motion, zoom), weather (snow, frost, fog, brightness), and digital artifacts (contrast, elastic transform, pixelate, JPEG compression). In our experiments, we specifically utilize severity level 5 for evaluation.

**ImageNet-V2 (Recht et al., 2019).** ImageNet-V2 is a set of re-created test sets for ImageNet-1k designed to assess model generalization under natural distribution shift. It replicates the original ImageNet data collection and annotation pipeline to curate new images for the same 1,000 classes, and provides three variants—matched-frequency, threshold-0.7, and top-images—each comprising 10,000 images (10 per class). The variants differ by selection criteria based on "selection frequency" (the fraction of annotators endorsing the target label): matched-frequency reproduces the selection-frequency distribution of the original validation set; threshold-0.7 retains images with selection frequency $\geq 0.7$; top-images uses the highest-agreement images.

**ImageNet-R (Hendrycks et al., 2021).** ImageNet-R (Renditions) is a benchmark for evaluating model robustness to non-photorealistic domain shifts. It comprises approximately 30,000 images collected from diverse artistic and abstract media—such as sketches, cartoons, paintings, graffiti, embroidery, sculptures and origami—mapped to a 200-class subset of ImageNet-1k. The renditions are intended to be label-preserving while inducing substantial shifts in texture, color, and style.

**ImageNet-Sketch (Wang et al., 2019).** ImageNet-Sketch is a benchmark for evaluating robustness and shape bias under domain shift. It comprises approximately 50,000 black-and-white line drawings mapped to the 1,000 ImageNet-1k classes. The sketches are intended to be label-preserving while largely removing texture cues, thereby emphasizing contour and global shape.

**DomainNet-126 (Peng et al., 2019).** DomainNet-126 is a large-scale multi-source domain adaptation benchmark constructed from a 126-class subset of the original DomainNet dataset. It contains images from four heterogeneous domains: clipart, painting, real, and sketch, covering both natural photographs and various non-photorealistic styles. These domains exhibit substantial variations in texture, color, abstraction level, and drawing style, making DomainNet-126 a challenging testbed for studying domain generalization and adaptation under significant appearance shifts.

Table 9: Performance comparison on ImageNet-C with ViT-Base model regarding **Accuracy** (%) under continual single-instance setting. The **bold** number indicates the best result.

| Method | Memory (MB) | Noise | | | Blur | | | | Weather | | | | Digital | | | | Average Acc. |
|---|---|---|---|---|---|---|---|---|---|---|---|---|---|---|---|---|---|
| | | Gauss. | Shot | Impl. | Defoc. | Glass | Motion | Zoom | Snow | Frost | Fog | Brit. | Contr. | Elas. | Pix. | JPEG | |
| No Adapt | - | 55.34 | 56.23 | 56.01 | 46.48 | 34.78 | 52.87 | 44.20 | 62.39 | 62.66 | 65.56 | 77.70 | 32.04 | 45.73 | 66.72 | 66.67 | 55.03 |
| ROID | 1345 | 0.10 | 0.08 | 0.08 | 0.08 | 0.08 | 0.08 | 0.08 | 0.08 | 0.08 | 0.08 | 0.08 | 0.08 | 0.08 | 0.08 | 0.08 | 0.08 |
| Yang et al. (2024) | 4033 | 3.86 | 6.68 | 1.82 | 0.13 | 9.16 | 8.10 | 3.77 | 0.82 | 0.34 | 0.14 | 79.24 | 0.10 | 0.21 | 73.60 | 70.95 | 17.26 |
| FOA | 702 | 6.94 | 1.74 | 1.42 | 1.72 | 0.40 | 0.48 | 0.68 | 0.86 | 0.72 | 0.98 | 2.04 | 0.88 | 0.86 | 0.22 | 0.96 | 1.39 |
| ZOA | 846 | 0.06 | 0.06 | 0.06 | 0.06 | 0.06 | 0.06 | 0.06 | 0.06 | 0.06 | 0.06 | 0.06 | 0.06 | 0.06 | 0.06 | 0.06 | 0.06 |
| T3A | 1128 | 55.18 | 56.72 | 56.00 | 38.88 | 32.96 | 50.96 | 42.82 | 60.14 | 60.18 | 64.22 | 76.48 | **40.24** | 43.12 | 66.60 | 68.48 | 54.20 |
| SAR | 1288 | 59.08 | 60.52 | 59.36 | 45.52 | **57.26** | **58.56** | **57.12** | 62.74 | 66.66 | 68.68 | 78.78 | 6.68 | **67.16** | 72.40 | **71.56** | 59.47 |
| TED | 696 | **61.78** | **62.40** | **62.98** | 51.82 | 39.50 | 57.74 | 47.44 | **68.92** | **68.52** | **68.72** | **80.80** | 31.74 | 56.20 | **70.30** | 67.94 | **59.79** |

Table 10: Performance comparison on ImageNet-C with ViT-Base model regarding **Accuracy** (%) under continual batch setting. The **bold** number indicates the best result.

| Method | Memory (MB) | Noise | | | Blur | | | | Weather | | | | Digital | | | | Average Acc. |
|---|---|---|---|---|---|---|---|---|---|---|---|---|---|---|---|---|---|
| | | Gauss. | Shot | Impl. | Defoc. | Glass | Motion | Zoom | Snow | Frost | Fog | Brit. | Contr. | Elas. | Pix. | JPEG | |
| No Adapt | - | 55.34 | 56.23 | 56.01 | 46.48 | 34.78 | 52.87 | 44.20 | 62.39 | 62.66 | 65.56 | 77.70 | 32.04 | 45.73 | 66.72 | 66.67 | 55.03 |
| ROID | 10866 | 60.68 | 63.04 | 63.00 | 56.60 | 55.84 | **62.72** | 58.86 | 66.78 | 66.70 | 70.76 | 79.24 | 12.14 | 57.02 | 69.46 | 71.04 | 60.93 |
| Yang et al. (2024) | 19489 | 58.52 | 63.92 | 64.46 | 50.34 | 40.66 | 56.58 | 44.94 | 67.62 | 68.42 | 68.48 | 77.74 | 39.54 | 51.66 | 68.46 | 69.70 | 59.27 |
| FOA | 1300 | 57.14 | 61.62 | 61.96 | **51.98** | **42.78** | 57.48 | **54.08** | 65.98 | 68.44 | 65.60 | 79.48 | **62.78** | 52.20 | 70.32 | **72.20** | **61.60** |
| ZOA | 1734 | 58.00 | 58.56 | 59.38 | 47.98 | 39.10 | 56.38 | 49.38 | 65.68 | 63.48 | 62.68 | 79.34 | 49.02 | 50.76 | 66.84 | 70.34 | 58.46 |
| T3A | 6818 | 58.70 | 61.18 | 61.40 | 47.48 | 48.88 | 58.38 | 50.92 | 62.82 | 62.30 | 64.90 | 78.32 | 60.40 | 54.78 | 67.50 | 69.44 | 60.49 |
| SAR | 1458 | 55.18 | 56.72 | 56.02 | 38.76 | 33.04 | 50.98 | 42.72 | 60.12 | 60.12 | 64.22 | 76.48 | 40.22 | 43.10 | 66.62 | 68.44 | 54.18 |
| BECoTTA | 3340MB | **61.82** | 62.52 | 62.27 | 51.64 | 38.10 | 58.16 | 48.65 | 67.79 | **68.82** | 36.77 | **82.20** | 30.69 | 49.38 | **71.99** | 72.11 | 57.53 |
| SURGEON | 2124MB | 60.43 | 60.95 | 62.88 | 50.39 | 36.66 | 56.52 | 47.35 | 68.32 | 67.32 | 69.61 | 80.91 | 32.93 | **57.94** | 70.42 | 70.53 | 59.54 |
| TED | 1290 | 61.38 | 61.98 | **62.64** | 51.08 | 39.16 | 57.70 | 47.20 | **68.76** | 68.60 | **71.50** | 79.96 | 30.78 | 55.78 | 69.28 | 67.08 | 59.53 |

**GSC-C.** GSC-C is a controlled corruption benchmark for keyword spotting that simulates everyday acoustic interference by mixing Google Speech Commands (GSC; Warden (2018)) with real-world background noise from ESC-50 (Piczak, 2015). We consider five noise categories—Animals, Natural, Human, Domestic, and Urban—and, within each, two representative soundscapes: dog, cat, pouring water, thunderstorm, crying baby, laughing, washing machine, vacuum cleaner, car horn, and fireworks. For each GSC utterance, we randomly sample a segment from an ESC-50 clip (to match the GSC duration) and additively mix it at diverse signal-to-noise ratios (SNRs), yielding multiple corrupted versions per utterance across SNR levels. Mixing is label-preserving and performed without time alignment beyond random cropping.

## C.2  More Details on Backbone

We use five backbone encoders: (1) ViT-Base (Dosovitskiy et al., 2021), (2) ResNet-50 (He et al., 2016), (3) EfficientNet-B0 (Tan & Le, 2019), (4) MobileNet-V4 (Qin et al., 2024), and (5) LSTM (Yang et al., 2025). Table 8 reports each model's hidden size, i.e., the dimensionality $D$ of the latent PC basis $\mathbf{V}_k$ used throughout the paper.

## C.3  More Ablation Studies

**Performance under Continual TTA Settings.** To demonstrate the practicality of TED in streaming scenarios, we extend our evaluation to continual adaptation, where the model adapts continuously to the test stream without resetting. To ensure a comprehensive comparison, we additionally incorporate recent state-of-the-art methods tailored for continual settings, including ROID (Marsden et al., 2024), ZOA (Deng et al., 2025), and the approach by Yang et al. (2024). We consider two settings: 1) **Continual single-instance** (Table 9): In this setting, standard gradient-based TTA methods often suffer from catastrophic forgetting due to repeated parameter updates on a long test sequence. In contrast, TED maintains a fixed backbone and exclusively optimizes an extremely low-dimensional latent coordinate for each sample. This design effectively prevents error accumulation, allowing TED to achieve state-of-the-art accuracy while requiring the lowest peak memory among all compared methods. 2) **Continual batch** (Table 10): For batch-level streaming (batch size = 64), we apply a shared shift vector $\mathbf{p}$ to all samples within a batch. While this global adjustment is inherently less granular than per-sample optimization, TED remains highly competitive in accuracy and retains its advantage of minimal memory footprint.

**Effect of Source Sample Size $N$ and Offline Nature.** First, we clarify that TED computes the latent basis $\mathbf{V}_k$ entirely **offline**. Similar to how BN statistics are frozen after training, $\mathbf{V}_k$ is pre-computed

Table 11: Performance comparison on ImageNet-C with ViT-Base model using different $N$ to obtain $\mathbf{V}_k$ regarding average **Accuracy** (%) and **Loss** value. The **bold** number indicates the best result.

| $N$ | 50k | 40k | 30k | 20k | 10k | 5k | 3k | 1k | 500 | 100 | 50 | 30 | 20 |
|---|---|---|---|---|---|---|---|---|---|---|---|---|---|
| Acc. | 57.82 | 57.69 | 57.28 | 56.89 | 56.53 | 55.94 | 53.41 | 51.52 | 54.75 | 58.66 | 59.08 | **59.14** | 59.13 |
| Loss | 2.33 | 2.45 | 2.52 | 2.59 | 2.43 | 2.62 | 2.77 | 2.75 | 2.38 | 2.20 | 2.23 | 2.25 | 2.19 |

Table 12: Performance comparison on ImageNet-C with ViT-Base model using different $N$ to obtain $\mathbf{V}_k$ regarding detailed **Accuracy** (%).

| $N$ | Noise | | | Blur | | | | Weather | | | | Digital | | | | Average |
| | Gauss. | Shot | Impl. | Defoc. | Glass | Motion | Zoom | Snow | Frost | Fog | Brit. | Contr. | Elas. | Pix. | JPEG | Acc. |
|---|---|---|---|---|---|---|---|---|---|---|---|---|---|---|---|---|
| 20 | 60.69 | 60.50 | 61.24 | 50.07 | 37.76 | 56.36 | 47.39 | 65.71 | 67.23 | 72.79 | 81.07 | 35.76 | 48.54 | 70.50 | 71.36 | 59.13 |
| 30 | 60.69 | 60.41 | 61.24 | 50.27 | 37.80 | 56.39 | 47.40 | 65.61 | 67.05 | 72.61 | 81.01 | 36.37 | 48.44 | 70.53 | 71.29 | 59.14 |
| 50 | 60.83 | 60.53 | 61.39 | 50.33 | 37.73 | 56.50 | 47.39 | 65.89 | 67.07 | 71.65 | 80.81 | 35.66 | 48.60 | 70.47 | 71.31 | 59.08 |
| 100 | 60.65 | 60.59 | 61.28 | 50.06 | 37.57 | 56.32 | 47.39 | 65.30 | 66.96 | 68.88 | 81.00 | 33.64 | 48.50 | 70.40 | 71.30 | 58.66 |
| 500 | 60.30 | 59.67 | 60.66 | 49.87 | 37.16 | 56.06 | 47.14 | 65.27 | 66.03 | 16.79 | 80.56 | 33.17 | 47.87 | 70.03 | 70.71 | 54.75 |
| 1000 | 59.22 | 59.11 | 59.82 | 48.99 | 37.19 | 55.46 | 46.76 | 64.73 | 65.31 | 7.26 | 80.21 | 1.02 | 48.10 | 69.28 | 70.31 | 51.52 |
| 3000 | 59.03 | 59.11 | 59.68 | 49.24 | 37.19 | 55.44 | 46.85 | 64.66 | 65.04 | 29.36 | 79.79 | 8.32 | 47.94 | 69.22 | 70.23 | 53.41 |
| 50k | 58.77 | 59.66 | 59.50 | 49.30 | 36.08 | 55.35 | 46.34 | 65.21 | 66.40 | 67.66 | 80.21 | 35.96 | 47.61 | 69.55 | 69.68 | 57.82 |

and stored (requiring negligible storage, $\approx 0.01\%$ of the model size for ViT-Base), ensuring no source data is needed during TTA.

Regarding the sensitivity to sample size, we extend our ablation to a wide range ($N \in [20, 50000]$) as shown in Table 11 and detailed Table 12. Contrary to the intuition that more data always yields better bases, we observe a *non-monotonic* behavior. Remarkably, extremely small sample sizes (e.g., $N = 20 \sim 50$) achieve an average accuracy of 59.14%, which is comparable to, or even superior to, utilizing the full validation set (57.82% at $N = 50$k). However, performance dips significantly in the medium regime ($N \approx$ 1k), hitting a local minimum of 51.5%. This phenomenon is most pronounced in corruptions like *Fog*, where accuracy starts high at small $N$ (72.8%), collapses at $N = 1$k (7.3%), and eventually recovers at $N = 50$k (67.7%).

We attribute this behavior to the purity of the subspace directions. With very few samples, the subspace captures only the most dominant, class-discriminative directions. As $N$ increases to the medium regime, the subspace begins to include less stable directions (noise) that vary across samples, which may mislead the unsupervised objective. When $N$ further increases, these unstable directions are statistically averaged out, restoring performance. This hypothesis is strongly corroborated by the interaction between $N$ and the subspace dimension $k$ (Table 13). For small $N$, performance is robust and insensitive to $k$, indicating the absence of noisy directions. In contrast, for medium $N$, accuracy drops sharply as $k$ increases, confirming that a larger dimension introduces more unstable components in this regime.

These findings underscore the practicality of TED: it remains highly effective even when only a handful of source samples are available.

**Effect of gradient free optimizer.** To validate the choice of CMA-ES, we compared it against several representative gradient-free baselines within the same latent subspace: Uniform Random Search, (1+1)-Evolution Strategy ((1+1)-ES), and Zeroth-Order SGD (ZO-SGD). As shown in Table 14, Uniform Random Search yields high instability and negligible gains due to the lack of directional guidance. While (1+1)-ES offers some improvement, it converges slowly, requiring significantly more iterations (e.g., 14 vs. 8 evaluations per sample on ViT) to achieve weaker TTA performance. We also observed that ZO-SGD suffers from high variance in gradient estimation within the single-instance regime, making it difficult to stabilize without extensive hyperparameter tuning. In contrast, CMA-ES consistently delivers reliable accuracy improvements with fewer evaluations and lower variance. Beyond algorithmic performance, CMA-ES is chosen for its practicality: it is fully gradient-free (relying solely on forward passes) and benefits from mature implementations in both Python and C/C++, facilitating seamless integration into edge-device runtimes. Thus, we adopt CMA-ES not as an algorithmic novelty, but as the most stable and hardware-friendly tool for our specific TTA formulation.

**Performance on in-distribution dataset.** We further evaluate TED's performance on in-distribution data (*i.e.*, the source test dataset). As shown in Table 15, our method achieves significant performance improvements across various models. This result highlights two key points: 1) The notable performance gain demonstrates that our method effectively mitigates catastrophic forgetting,

Table 13: **Accuracy** (%) on ImageNet-C with ViT-Base using diverse $N$ and $k$.

| $N \setminus k$ | 4 | 8 | 16 | 32 |
|---|---|---|---|---|
| 30 | 58.03 | 59.09 | 59.14 | 59.15 |
| 1k | 54.63 | 56.25 | 51.52 | 48.74 |
| 5k | 55.39 | 56.61 | 57.82 | 57.90 |

Table 14: Performance comparison on ImageNet-C with ViT-Base model using different gradient-free optimizers regarding **Accuracy** (%).

| Optimizer | Noise | | | Blur | | | | | Weather | | | | Digital | | | Average |
|---|---|---|---|---|---|---|---|---|---|---|---|---|---|---|---|---|
| | Gauss. | Shot | Impl. | Defoc. | Glass | Motion | Zoom | Snow | Frost | Fog | Brit. | Contr. | Elas. | Pix. | JPEG | Acc. |
| Uniform Random Search | 43.86 | 46.68 | 41.82 | 40.13 | 49.16 | 48.10 | 43.77 | 40.82 | 40.34 | 40.14 | 63.24 | 40.10 | 40.21 | 63.60 | 60.95 | 46.86 |
| (1+1) ES | 56.10 | 56.93 | 56.79 | 47.20 | 35.17 | 53.49 | 44.82 | 62.95 | 63.59 | 66.94 | 78.10 | 33.11 | 46.22 | 67.26 | 67.23 | 55.73 |
| ZO-SGD | 55.06 | 55.88 | 52.62 | 45.53 | 34.41 | 52.50 | 43.96 | 62.39 | 59.66 | 64.82 | 77.59 | 21.29 | 45.46 | 66.48 | 66.54 | 53.61 |
| CMA-ES | 58.77 | 59.66 | 59.50 | 49.30 | 36.08 | 55.35 | 46.34 | 65.21 | 66.40 | 67.66 | 80.21 | 35.96 | 47.61 | 69.55 | 69.68 | 57.82 |

as it even enhances the model's performance on the original data distribution. 2) The improvement can be attributed to the inherent distribution shift between the source test data and the training data. Our TED framework adjusts the latent representations of test samples to be more compactly aligned within the defined principal subspace, which reduces uncertainty and enables the model to produce more confident predictions.

**Performance on DomainNet-126.** To further assess the robustness of TED under rigorous conditions beyond synthetic corruptions (e.g., ImageNet-C), we extend our evaluation to DomainNet-126 (Peng et al., 2019). Unlike corruption benchmarks that primarily introduce texture or noise degradations while preserving object geometry, DomainNet features significant semantic and stylistic variations across four distinct domains (Real, Sketch, Clipart, and Painting), presenting a substantial challenge for adaptation methods.

As presented in Table 16, TED demonstrates superior generalization capabilities across these severe distribution shifts. In the online batch setting, TED remains highly competitive with state-of-the-art methods. However, the advantage of TED becomes most pronounced in the single-instance setting. Due to the extreme diversity and large domain gaps inherent in DomainNet, existing baselines are prone to error accumulation and catastrophic forgetting when processing samples sequentially. In contrast, TED effectively mitigates these issues, maintaining stable and robust performance. These results confirm that TED is not only effective against local corruptions but also resilient to complex structural domain shifts.

**Quantization of TED.** Table 17, Table 18 and Table 19 demonstrates the performance of TED under various quantization configurations. For QTED-V1, quantizing $\mathbf{p}$ into a 1-bit representation reduces the optimization process to controlling $k$ binary switches, significantly lowering hardware costs while achieving an average accuracy of 57.63% and 62.52% on ImageNet series, close to the original TED. With $k$ fixed at 2 for the KWS task, $\mathbf{p}$ is naturally quantized to 1-bit; consequently, TED specializes to QTED-V1. Since For QTED-V2, fixed-point arithmetic is used to simulate CMA-ES. Among these, QTED-V2 (8b4) achieves 56.32% and 61.88% accuracy in the IC task, demonstrating that 8-bit fixed-point arithmetic is sufficient for effective optimization. In the KWS task with simpler model architecture, 4-bit QTED-V2 is good enough for effective TTA. These results confirm the feasibility of our methods for resource-limited edge devices, with QTED-V1 minimizing resource overhead and QTED-V2 ensuring compatibility with fixed-point hardware.

## THE USE OF LARGE LANGUAGE MODELS

The manuscript benefited from language polishing suggestions provided by large language models. All scientific content remains the authors' responsibility.

Table 15: Performance on in-distribution dataset with different models regarding **Accuracy** (%).

| Model | Vit-Base | ResNet-50 | EfficientNet-B0 | MobileNet-V4 |
|---|---|---|---|---|
| No Adapt | 85.16 | 70.74 | 78.53 | 71.04 |
| TED | 87.05 | 76.71 | 85.31 | 79.91 |
| **Improvement** | **+1.89** | **+5.97** | **+6.78** | **+8.87** |

Table 16: Performance Comparison on DomainNet-126 with ResNet-50 model regarding **Accuracy** (%). **BS** stands for batch size.

(a) `Painting` as Source Domain

| Method | Real | Sketch | Clipart | Avg. |
|---|---|---|---|---|
| No Adapt | 74.84 | 49.70 | 53.26 | 59.27 |
| SAR (BS=64) | 73.58 | 53.96 | 53.50 | 60.35 |
| ROID (BS=64) | 75.04 | 57.30 | 57.24 | 63.19 |
| SAR (BS=1) | 6.42 | 4.24 | 3.22 | 4.63 |
| ROID (BS=1) | 0.48 | 0.16 | 0.16 | 0.27 |
| MEMO (BS=1) | 40.38 | 17.82 | 25.42 | 27.87 |
| TED (BS=64) | 78.24 | 53.74 | 55.50 | 62.49 |
| TED (BS=1) | 76.08 | 52.94 | 54.22 | 61.08 |

(b) `Clipart` as Source Domain

| Method | Real | Sketch | Clipart | Avg. |
|---|---|---|---|---|
| No Adapt | 46.16 | 60.50 | 43.84 | 50.17 |
| SAR (BS=64) | 48.80 | 64.36 | 47.34 | 53.50 |
| ROID (BS=64) | 51.24 | 64.64 | 49.14 | 55.01 |
| SAR (BS=1) | 4.98 | 5.80 | 4.38 | 5.05 |
| ROID (BS=1) | 0.16 | 0.48 | 0.30 | 0.31 |
| MEMO (BS=1) | 40.38 | 15.48 | 27.92 | 18.77 |
| TED (BS=64) | 47.24 | 63.24 | 46.08 | 52.19 |
| TED (BS=1) | 47.20 | 62.44 | 44.74 | 51.46 |

(c) `Real` as Source Domain

| Method | Real | Sketch | Clipart | Avg. |
|---|---|---|---|---|
| No Adapt | 48.48 | 54.88 | 59.32 | 54.23 |
| SAR (BS=64) | 57.56 | 58.98 | 67.92 | 61.49 |
| ROID (BS=64) | 58.76 | 61.44 | 68.58 | 62.93 |
| SAR (BS=1) | 4.78 | 4.50 | 5.26 | 4.85 |
| ROID (BS=1) | 0.30 | 0.16 | 0.48 | 0.31 |
| MEMO (BS=1) | 13.92 | 26.00 | 21.68 | 20.53 |
| TED (BS=64) | 56.52 | 61.32 | 64.46 | 60.77 |
| TED (BS=1) | 54.44 | 59.50 | 63.50 | 59.15 |

(d) `Sketch` as Source Domain

| Method | Real | Sketch | Clipart | Avg. |
|---|---|---|---|---|
| No Adapt | 55.62 | 61.88 | 47.84 | 55.11 |
| SAR (BS=64) | 54.36 | 62.90 | 48.40 | 55.22 |
| ROID (BS=64) | 57.48 | 65.18 | 52.66 | 58.44 |
| SAR (BS=1) | 3.74 | 4.66 | 3.58 | 3.99 |
| ROID (BS=1) | 0.16 | 0.30 | 0.16 | 0.21 |
| MEMO (BS=1) | 40.38 | 26.12 | 22.92 | 21.63 |
| TED (BS=64) | 56.40 | 63.28 | 49.82 | 56.50 |
| TED (BS=1) | 56.50 | 63.12 | 49.16 | 56.26 |

Table 17: Performance of QTED on ImageNet-C with ViT-Base model regarding **Accuracy** (%). QTED-V2 ($x$b$y$) indicates CMA-ES using $x$-bit fixed point with $y$-bit integer.

| Method | Noise Gauss. | Shot | Impl. | Defoc. | Blur Glass | Motion | Zoom | Snow | Weather Frost | Fog | Brit. | Contr. | Digital Elas. | Pix. | JPEG | Average Acc. |
|---|---|---|---|---|---|---|---|---|---|---|---|---|---|---|---|---|
| TED | 58.77 | 59.66 | 59.50 | 49.30 | 36.08 | 55.35 | 46.34 | 65.21 | 66.40 | 67.66 | 80.21 | 35.96 | 47.61 | 69.55 | 69.68 | 57.82 |
| QTED-V1 | 59.41 | 60.15 | 60.09 | 49.86 | 36.48 | 55.94 | 46.70 | 65.60 | 66.74 | 60.90 | 80.46 | 34.55 | 47.72 | 69.83 | 70.01 | 57.63 |
| QTED-V2 (8b5) | 56.16 | 57.11 | 56.93 | 47.17 | 35.14 | 53.60 | 44.76 | 63.18 | 63.94 | 67.60 | 78.37 | 33.03 | 46.26 | 67.48 | 67.49 | 55.88 |
| QTED-V2 (8b4) | 56.75 | 57.75 | 57.35 | 47.63 | 35.32 | 53.99 | 45.12 | 63.60 | 64.39 | 68.34 | 78.66 | 33.71 | 46.51 | 67.83 | 67.88 | 56.32 |
| QTED-V2 (8b3) | 56.73 | 57.61 | 57.42 | 47.63 | 35.35 | 53.90 | 45.04 | 63.49 | 64.35 | 68.07 | 78.65 | 33.16 | 46.53 | 67.75 | 67.77 | 56.23 |
| QTED-V2 (8b2) | 56.64 | 57.59 | 57.37 | 47.55 | 35.37 | 53.99 | 45.08 | 63.52 | 64.28 | 67.76 | 78.56 | 33.41 | 46.49 | 67.72 | 67.81 | 56.21 |
| QTED-V2 (4b4) | 55.06 | 55.88 | 55.62 | 45.53 | 34.41 | 52.50 | 43.96 | 62.39 | 61.66 | 64.82 | 77.59 | 33.29 | 45.46 | 66.48 | 66.54 | 54.75 |
| QTED-V2 (4b2) | 55.41 | 56.35 | 56.15 | 46.56 | 34.78 | 52.93 | 44.24 | 62.44 | 62.85 | 65.84 | 77.82 | 32.10 | 45.78 | 66.76 | 66.72 | 55.12 |

Table 18: Performance of QTED on ImageNet-V2/R/Sketch with ViT-Base model regarding **Accuracy** (%). QTED-V2 ($x$b$y$) indicates CMA-ES using $x$-bit fixed point with $y$-bit integer.

| Method | Accuracy (%) V2 | R | Sketch | Avg. |
|---|---|---|---|---|
| TED | 78.15 | 65.29 | 47.73 | 63.72 |
| QTED-V1 | 77.46 | 63.31 | 46.79 | 62.52 |
| QTED-V2 (8b5) | 76.94 | 62.02 | 46.33 | 61.76 |
| QTED-V2 (8b4) | 77.21 | 62.06 | 46.37 | 61.88 |
| QTED-V2 (8b3) | 76.86 | 61.45 | 45.99 | 61.43 |
| QTED-V2 (8b2) | 76.26 | 60.42 | 45.39 | 60.69 |
| QTED-V2 (4b4) | 75.17 | 57.70 | 44.65 | 59.17 |
| QTED-V2 (4b2) | 75.46 | 59.42 | 44.88 | 59.92 |

Table 19: Performance of QTED on GSC-C with LSTM model regarding **Accuracy** (%). QTED-V2 ($x$b$y$) indicates CMA-ES using $x$-bit fixed point with $y$-bit integer.

| SNR | Method | Animals | | Natural | | Human | | Domestic | | Urban | | Average |
| | | dog | cat | pouringwater | thunderstorm | cryingbaby | laughing | washingmachine | vacuumcleaner | carhorn | fireworks | Acc. |
| --- | --- | --- | --- | --- | --- | --- | --- | --- | --- | --- | --- | --- |
| -10 dB | TED (QTED-V1) | 64.25 | 63.58 | 59.73 | 66.47 | 61.99 | 61.94 | 59.46 | 56.98 | 59.32 | 64.90 | 61.86 |
| | QTED-V2 (8b2) | 63.70 | 62.65 | 57.71 | 66.63 | 60.56 | 60.74 | 57.09 | 55.03 | 58.37 | 63.42 | 60.59 |
| | QTED-V2 (8b1) | 63.85 | 62.85 | 57.33 | 66.67 | 60.67 | 60.43 | 58.01 | 54.11 | 58.53 | 63.60 | 60.61 |
| | QTED-V2 (4b2) | 63.65 | 63.17 | 57.26 | 66.65 | 60.64 | 60.51 | 58.05 | 53.87 | 58.68 | 63.66 | 60.61 |
| | QTED-V2 (4b1) | 63.42 | 62.10 | 56.02 | 66.69 | 59.69 | 59.50 | 55.51 | 52.26 | 57.66 | 62.19 | 59.50 |
| -15 dB | TED (QTED-V1) | 60.84 | 57.99 | 57.71 | 62.28 | 58.04 | 58.98 | 57.83 | 55.38 | 56.00 | 60.41 | 58.55 |
| | QTED-V2 (8b2) | 59.66 | 55.90 | 54.82 | 62.14 | 56.39 | 56.97 | 55.80 | 53.22 | 54.51 | 58.29 | 56.77 |
| | QTED-V2 (8b1) | 59.22 | 56.88 | 53.69 | 62.00 | 56.55 | 55.76 | 55.48 | 51.63 | 54.37 | 57.37 | 56.30 |
| | QTED-V2 (4b2) | 59.26 | 56.85 | 53.77 | 61.98 | 56.59 | 55.79 | 55.40 | 51.53 | 54.41 | 57.09 | 56.27 |
| | QTED-V2 (4b1) | 58.27 | 55.09 | 51.57 | 61.97 | 54.90 | 54.31 | 51.24 | 49.47 | 53.18 | 56.08 | 54.61 |
| -20 dB | TED (QTED-V1) | 59.07 | 54.71 | 57.20 | 59.35 | 56.94 | 57.94 | 58.22 | 55.54 | 54.50 | 58.07 | 57.15 |
| | QTED-V2 (8b2) | 56.98 | 51.06 | 53.86 | 59.36 | 55.39 | 54.59 | 56.62 | 53.18 | 52.25 | 55.90 | 54.92 |
| | QTED-V2 (8b1) | 55.99 | 52.12 | 51.58 | 59.23 | 55.13 | 52.36 | 54.03 | 52.50 | 51.69 | 53.54 | 53.82 |
| | QTED-V2 (4b2) | 56.07 | 51.97 | 51.78 | 59.05 | 55.23 | 52.51 | 53.90 | 52.42 | 51.79 | 53.60 | 53.83 |
| | QTED-V2 (4b1) | 54.41 | 49.80 | 48.61 | 58.74 | 53.24 | 50.34 | 48.63 | 49.35 | 50.58 | 51.79 | 51.55 |

