# OpenReview forum: "Efficient Edge Test-Time Adaptation via Latent Feature Coordinate Correction"
_ICLR.cc/2026/Conference — Submitted to ICLR 2026_

### Official Review · Reviewer_JVp9 · 2025-10-28

**Soundness:** 3
**Presentation:** 3
**Contribution:** 4
**Rating:** 6
**Confidence:** 5

**Summary:**

This paper highlights the importance of computational efficiency in test-time adaptation (TTA) and proposes an efficient framework called TED (**T**TA for **E**dge **D**evices). In brief, TED first precomputes basis vectors from source data during the source training phase. At test time, the encoded representation of a target sample is optimized through the Covariance Matrix Adaptation Evolution Strategy (CMA-ES) [a] to align it with the source distribution, and the resulting projected representation is then used for the final prediction. The proposed approach is extensively evaluated across multiple datasets, tasks, devices, and network architectures, demonstrating its scalability.

[a] The CMA Evolution Strategy: A Tutorial. arXiv 2016

**Strengths:**

1. The method is novel and described clearly. Figure 2 effectively summarizes the overall procedure.
2. The effectiveness of the proposed method is shown in both performance and resource efficiency across various architectures and edge devices. It is surprising that TED outperforms T3A with about one-third of the time cost
3. The experiments cover multiple benchmarks, including image classification and speech recognition tasks.

**Weaknesses:**

1. Although the authors emphasize TTA efficiency, the discussion overlooks recent studies with similar goals. The related work (eg, T3A, MEMO, and SAR) mainly covers older methods. Adding comparisons or discussions (eg, Line 59) with recent efficiency-focused works [1–4], and including time and memory results beyond T3A, MEMO, and SAR, would strengthen the paper.
2. The proposed method shows strong results, but it is questionable whether this advantage mainly comes from the single-instance setup, where the baselines naturally struggle. Although SAR and MEMO follow the same condition, they are relatively simple and outdated. It would be valuable to evaluate TED in batch adaptation settings and in comparison with existing works such as T3A, MEMO, and [1–4]. (In fact, the single-instance setup in this paper cannot be considered fully practical, similar to the “multiple images in a batch” setup, since the inputs are not streaming or continuous [5].)
3. All six subplots in Figure 3 convey the same message: "the adapted features are closer to the original features than the corrupted ones". Since this information is redundant, it might be better to keep only Figure 3(f). In addition, Table 8 in the supplementary material would fit better in the main paper.

[1] MECTA: Memory-Economic Continual Test-Time Model Adaptation ICLR 2023

[2] EcoTTA: Memory-Efficient Continual Test-time Adaptation via Self-distilled Regularization CVPR 2023

[3] BECoTTA: Input-dependent Online Blending of Experts for Continual Test-time Adaptation ICML 2024

[4] SURGEON: Memory-Adaptive Fully Test-Time Adaptation via Dynamic Activation Sparsity CVPR 2025

[5] NOTE: Robust Continual Test-time Adaptation Against Temporal Correlation NIPS 2022

**Questions:**

1. Please refer to the weaknesses above.
2. The name “TED” does not clearly represent the proposed approach. Revising it to better align with the proposed approach could be considered.

---

> ### Author Response · Authors · 2025-11-20
> **Thank you; Address your concerns [1/2]**
>
> Dear Reviewer,
>
> Thank you for your thoughtful comments, which have helped us make our work clearer and more solid. We have carefully addressed all your concerns in detail. In the revised manuscript, changes are highlighted in purple and yellow, with yellow indicating changes also requested by other reviewers.
>
> **[W1] Comparison with additional baselines.**
>
> We thank the reviewer for pointing out these recent efficiency‑oriented TTA methods. We have updated the related‑work section (around Line 59) to discuss them more explicitly and added corresponding experiments.
>
> 1. MECTA and EcoTTA primarily target models with BN layers. Their update rules operate on BN statistics, which limits applicability to architectures such as ViTs (with LayerNorm) and standard LSTMs (without normalization). To still provide a direct comparison, we evaluate them on ResNet‑50, where BN is available, and report both accuracy and efficiency.
>
> 2. Methods such as BECoTTA and SURGEON have a broader applicability and are therefore included as main baselines in our experiments. Compared to CoTTA, they substantially reduce the number of updated parameters, but they still **rely on back‑propagation‑based updates**. In contrast, TED only optimizes a k‑dimensional latent vector via forward passes, which leads to even lower computational and memory overhead.
>
> **T1. Performance Comparison on ImageNet-C with ResNet50.**
> |Method|Memory|Noise|||Blur||||Weather||||Digital||||Acc.|
> |-|-|-|-|-|-|-|-|-|-|-|-|-|-|-|-|-|-|
> |||**Gauss.**|**Shot**|**Impl.**|**Defoc.**|**Glass**|**Motion**|**Zoom**|**Snow**|**Frost**|**Fog**|**Brit.**|**Contr.**|**Elas.**|**Pix.**|**JPEG**|**Avg.**|
> |No&nbsp;Adapt||4.47|4.74|4.06|8.11|5.96|9.59|14.74|6.92|14.47|12.32|45.80|0.72|11.08|18.19|32.54|12.91|
> |MECTA|674MB|3.53|3.92|3.14|9.03|5.97|11.16|15.91|6.02|14.17|12.74|50.56|0.07|11.50|20.44|36.54|13.64|
> |EcoTTA|948MB|4.00|4.47|3.65|9.77|5.22|13.50|15.79|7.30|10.65|10.83|55.61|1.70|14.26|24.15|41.61|14.83|
> |TED|452MB|4.99|5.18|4.41|11.36|7.32|14.26|18.29|7.00|15.32|15.01|55.32|0.25|13.07|23.29|40.97|15.74|
>
> **T2. Performance Comparison on ImageNet-C with ViT-Base.**
> |Method|Memory|Noise|||Blur||||Weather||||Digital||||Acc.|
> |-|-|-|-|-|-|-|-|-|-|-|-|-|-|-|-|-|-|
> |||**Gauss.**|**Shot**|**Impl.**|**Defoc.**|**Glass**|**Motion**|**Zoom**|**Snow**|**Frost**|**Fog**|**Brit.**|**Contr.**|**Elas.**|**Pix.**|**JPEG**|**Avg.**|
> |No&nbsp;Adapt||55.34|56.23|56.01|46.48|34.78|52.87|44.20|62.39|62.66|65.56|77.70|32.04|45.73|66.72|66.67|55.03|
> |BECoTTA|778MB|55.67|56.45|56.29|46.21|33.68|52.66|43.67|62.20|63.37|68.25|77.58|33.74|45.09|66.70|66.78|55.22|
> |SURGEON|716MB|58.70|59.22|59.23|48.82|35.29|55.06|45.87|64.83|65.94|61.76|79.56|34.46|46.90|69.02|69.36|56.93|
> |TED|452MB|58.77|59.66|59.50|49.30|36.08|55.35|46.34|65.21|66.40|67.66|80.21|35.96|47.61|69.55|69.68|57.82|
>
> **[W2] Performance on batch setting.**
>
> Thanks for your advice. Together with reviewer 8Tmh’s comments, we additionally evaluate TED under continual batch settings (batch size = 64) without resetting the model between test samples or domains. TED is designed for per-sample coordinate correction, so for a batch we apply a shared shift vector *p* to all samples in the batch. This global shift is less precise than per-sample optimization, yet TED remains highly competitive in terms of accuracy and still exhibits the lowest peak memory footprint in this streaming batch setting.
>
> **T3. Continual batch setting. Evaluated on ImageNet-C with ViT-Base.**
> |Method|Memory||Noise|||Blur||||Weather||||Digital|||Acc.
> |-|-|-|-|-|-|-|-|-|-|-|-|-|-|-|-|-|-|
> |||**Gauss.**|**Shot**|**Impl.**|**Defoc.**|**Glass**|**Motion**|**Zoom**|**Snow**|**Frost**|**Fog**|**Brit.**|**Contr.**|**Elas.**|**Pix.**|**JPEG**|**Avg.**|
> |No&nbsp;Adapt||55.34|56.23|56.01|46.48|34.78|52.87|44.20|62.39|62.66|65.56|77.70|32.04|45.73|66.72|66.67|55.03|
> |ROID|10866MB|60.68|63.04|63.00|56.60|55.84|62.72|58.86|66.78|66.70|70.76|79.24|12.14|57.02|69.46|71.04|60.93|
> |CVPR'24|19489MB|58.52|63.92|64.46|50.34|40.66|56.58|44.94|67.62|68.42|66.48|77.74|39.54|51.66|68.46|69.70|59.27|
> |FOA|1300MB|57.14|61.62|61.96|51.98|42.78|57.48|54.08|65.98|68.44|65.60|79.48|62.78|52.20|70.32|72.20|61.60|
> |ZOA|1734MB|58.00|58.56|59.38|47.98|39.10|56.38|49.38|65.68|63.48|62.68|79.34|49.02|50.76|66.84|70.34|58.46|
> |T3A|6818MB|58.70|61.18|61.40|47.48|48.88|58.38|50.92|62.82|62.30|64.90|78.32|60.40|54.78|67.50|69.44|60.49|
> |SAR|1458MB|55.18|56.72|56.02|38.76|33.04|50.98|42.72|60.12|60.12|64.22|76.48|40.22|43.10|66.62|68.44|54.18|
> |BECoTTA|3340MB|61.82|62.52|62.27|51.64|38.10|58.16|48.65|67.79|68.82|36.77|82.20|30.69|49.38|71.99|72.11|57.53|
> |SURGEON|2124MB|60.43|60.95|62.88|50.39|36.66|56.52|47.35|68.32|67.32|69.61|80.91|32.93|57.94|70.42|70.53|59.54|
> |TED|1290MB|61.38|61.98|62.64|51.08|39.16|57.70|47.20|68.76|68.60|71.50|79.96|30.78|55.78|69.28|67.08|59.53|

---

> ### Author Response · Authors · 2025-11-20
> **Thank you; Address your concerns [2/2]**
>
> **[W3] Concerns on Figure 3 and Table 8.**
>
> Thanks for the suggestion. In the revised version, we keep only Fig. 3(f) to summarize the feature‑alignment effect and move the original Table 8 from the supplementary to the main paper, as recommended.
>
> **[Q2] Name of our method.**
>
> We appreciate your suggestion. Since the paper is currently under review and all experiments, tables, and figures consistently use the name “TED”, we prefer to keep this name during the rebuttal phase to avoid confusion among reviewers. We will, however, revise the name in subsequent version to better reflect the core idea of our approach.

---

> ### Comment · Reviewer_JVp9 · 2025-11-26
>
> Thank you for providing a thorough response. I have checked all author responses. The clarification for [C1, C2, C6, C7, C8] is reasonable to me. The experimental setup and scope of this work are understandable from a research perspective, while some concerns suggest that further engineering effort may be beneficial [C3, C4, C5].
>
> Furthermore, I am especially satisfied with the motivation, pursuing a parameter-updating-free approach represents an important step toward more reliable TTA. While the proposed method builds on known components such as PCA and CMA-ES, I believe that demonstrating their impact within the TTA framework constitutes a meaningful contribution.
>
> I appreciate the authors’ clarifications and additional experiments [E1–E6], which address my concerns. Accordingly, **I am raising my score**. However, I respect the assessments by the other reviewers, and I remain open to additional discussion if appropriate.
>
> As a further suggestion, it would be helpful if the authors could include an introduction to the newly added baselines (e.g., TTN, SAR, EcoTTA, and MECTA) and provide relevant implementation details in the supplementary material.
>
> Clarifications
>
> - [C1] Motivation and practicality of the strict single-instance setting
> - [C2] Source-domain sample requirement.
> - [C3] Source statistics and its sensitivity.
> - [C4] Potentially unfair comparisons.
> - [C5] Concerns on baselines.
> - [C6] Technical novelty.
> - [C7] Concerns on Figure 3 and Table 8.
> - [C8] Name of our method. (this reviewer's minior concern)
>
> Experiments
>
> - [E1] Performance comparison on ImageNet-C with ViT-Base model using different N
> - [E2]  Accuracy on ImageNet‑C vs. source sample count *N* and subspace dimension *k*
> - [E3] Completeness of baselines (including TTN and SAR) across architectures and devices.
> - [E4] Comparison with additional baselines (including EcoTTA and MECTA)
> - [E5] Performance on batch setting.
> - [E6] Dataset diversity (DomainNet-126)

---

> > ### Author Response · Authors · 2025-11-28
> > **Thank you and further response**
> >
> > Dear Reviewer,
> >
> > We sincerely thank the reviewer for the positive feedback and the decision to raise the score. We are particularly encouraged by your recognition of our core motivation that pursuing a **parameter-updating-free** approach as a step toward more reliable and deployable TTA.
> >
> > Regarding your final suggestion: We wholeheartedly agree that providing clear context for the added baselines is essential and we will include it in the final version.
> >
> > Your constructive comments throughout this process have significantly strengthened both the technical depth and the presentation of our work. Thank you for your support.
> >
> > Sincerely,
> >
> > The Authors

---

### Official Review · Reviewer_QpSX · 2025-10-29

**Soundness:** 3
**Presentation:** 3
**Contribution:** 2
**Rating:** 2
**Confidence:** 5

**Summary:**

Paper Summary: This paper proposes TED (Test-time adaptation for Edge Devices), a novel single-instance test-time adaptation method designed for resource-constrained edge devices. The method performs forward-only coordinate optimization in the principal subspace of latent features using CMA-ES (Covariance Matrix Adaptation Evolution Strategy). By updating only a compact low-dimensional vector representing coordinates in the source latent principal subspace, TED achieves efficient adaptation without backpropagation while keeping model parameters frozen. The authors validate their approach on image classification (ImageNet series) and keyword spotting (GSC-C) tasks, demonstrating up to 63× reduction in computational complexity while achieving state-of-the-art performance in single-instance TTA scenarios.

**Strengths:**

Paper Strengths: (1) The problem formulation is well-motivated and addresses a critical gap in TTA for edge devices, where gradient-based and batch-dependent methods are impractical due to computational and memory constraints.
(2) The core idea of correcting coordinates in the latent principal subspace is elegant and theoretically grounded. The mathematical framework (Equations 2-7) clearly demonstrates how the proposed method reformulates TTA as a coordinate correction problem.
(3) The method is architecture-agnostic and demonstrates strong versatility across different backbone networks and diverse tasks (image classification and keyword spotting).

**Weaknesses:**

Paper Weaknesses: (1) The overall novelty appears limited. While the overall framework is well-designed, the individual components show limited innovation. The use of PCA-based subspace projection is standard in dimensionality reduction, and CMA-ES is an existing optimization algorithm. The main contribution appears to be the combination of these techniques rather than fundamental algorithmic innovation.
(2) In Section 3.2 and Appendix A, the authors claim that minimizing Shannon entropy drives OOD features closer to the source domain in Vk. However, this connection relies on strong assumptions that may not hold in practice, especially under severe distribution shifts. The paper does not empirically validate these assumptions or provide theoretical guarantees.
(3) The paper does not adequately explain why CMA-ES is the optimal choice for this problem. While CMA-ES is gradient-free, other evolutionary algorithms or simpler random search methods might be more efficient for low-dimensional optimization. No comparison with alternative gradient-free optimizers is provided.
(4) As shown in Table 1, TED shows minimal improvements for certain corruption types. The paper acknowledges this limitation only briefly in Section 4.3 but does not provide sufficient analysis of when and why the method fails.
(5) Although the experimental results are generally honest, the datasets used are limited. It is recommended that the authors conduct further validation on more challenging datasets (e.g., CIFAR10-C, CIFAR100-C and DomainNet) to enhance the generalizability and persuasiveness of the conclusions.
(6) The paper does not include comparisons with recent state-of-the-art methods such as [1] and [2], which achieve good performance. As these approaches are both effective and relatively simple, their omission makes it difficult to assess the novelty and practical advantage of the proposed approach.
[1] A Versatile Framework for Continual Test-Time Domain Adaptation: Balancing Discriminability and Generalizability, CVPR 2024
[2] Universal Test-time Adaptation through Weight Ensembling, Diversity Weighting, and Prior Correction, WACV 2024

**Questions:**

See Weaknesses.

---

> ### Author Response · Authors · 2025-11-20
> **Thank you; Address your concerns [1/4]**
>
> Dear Reviewer,
>
> Thank you for your thoughtful comments, which have helped us make our work clearer and more solid. We have carefully addressed all your concerns in detail. In the revised manuscript, changes are highlighted in red and yellow, with yellow indicating changes also requested by other reviewers.
>
> **[W1] Technical novelty.**
>
> We agree that PCA and CMA‑ES are standard techniques, and we do not claim novelty on these tools themselves. The novelty of our work lies in a **non‑trivial TTA paradigm**: under strict edge constraints, we show that it is sufficient to adapt only **a handful of latent scalars** per sample ($16$ parameters for ViT, $2$ for LSTM) while keeping the entire backbone frozen.
>
> This leads to properties that standard “PCA + optimizer” combinations do not offer:
> 1. **Extreme parameter and memory efficiency.** The adaptable state is a fixed‑size latent vector whose dimension is independent of the backbone size, which makes the method highly hardware‑friendly and easy to deploy on constrained devices.
> 2. **Stability even under single‑instance or continual updates.** Because we only update a tiny latent vector and never touch backbone weights, our method naturally avoids catastrophic forgetting and instability that typically arise when repeatedly adapting large parameter sets or normalization layers in edge scenarios.
>
> Despite this **ultra‑low‑dimensional update**, the framework consistently provides strong TTA across **multiple architectures** (rather than being limited to models with BN layers or prompt‑based designs) and **application domains**, suggesting that the formulation itself is non‑trivial rather than a superficial combination of existing components. We will clarify in the revision that our contribution is this **edge‑oriented, minimal‑state latent adaptation paradigm**, not new PCA or optimization algorithms.
>
> **[W2] Entropy minimization, assumptions, and empirical justification**
>
> Our argument in Sec. 3.2 and Appendix A is meant to explain **why** entropy minimization in $\mathbf V_k$ is a reasonable mechanism, rather than to claim an unconditional theoretical guarantee. It explicitly relies on **standard modeling assumptions** for deep classifiers on the source domain—softmax calibration and approximately shared‑covariance class‑conditional structure in the principal subspace [1-3]. Under these assumptions, minimizing Shannon entropy within the fixed source subspace $\mathbf V_k$ encourages features to move toward high‑density regions around source class prototypes. In the revision we will make these assumptions explicit in the main text and avoid wording that suggests a universal guarantee under arbitrarily severe shifts.
>
> Importantly, we **do empirically validate** the effect of the entropy objective on the latent geometry. In Fig. X, we analyze 1,000 ImageNet‑C Gaussian Noise samples and their clean counterparts with ViT‑B, and show that after TED adaptation:
> (i) the means and standard deviations of the top 50 latent dimensions align much more closely with those of clean features;
> (ii) the distance to clean features decreases (Euclidean distance $31.1 \to 26.4$) while cosine similarity increases ($0.54 \to 0.58$).
>
> Thus, both the standard assumptions we adopt and the above empirical evidence support our claim that **subspace restriction + entropy minimization** tends to steer OOD features closer to the source feature structure in $\mathbf V_k$, which explains the effectiveness of our TTA procedure in practice.

---

> ### Author Response · Authors · 2025-11-20
> **Thank you; Address your concerns [2/4]**
>
> **[W3] On the choice of CMA‑ES**
>
> Thanks for your advice. We compared CMA‑ES with several representative gradient‑free baselines in the same latent space. Uniform random search lacks directional guidance and shows highly unstable behaviour across runs, while a simple (1+1)-ES needs significantly more iterations (e.g., ≥14 vs. 8 evaluations per sample on ViT) yet still yields weaker TTA gains. We also tested a zeroth‑order SGD (ZO) variant, whose finite‑difference gradients exhibit very high variance in the single‑instance regime, making it hard to stabilize without heavy tuning. Among them, CMA‑ES consistently provided more reliable improvements with **fewer evaluations and lower variance**.
>
> **T1. TED performance comparison with diver gradient-free optimization method.**
> |Method|Noise|||Blur||||Weather||||Digital||||Acc.|
> |-|-|-|-|-|-|-|-|-|-|-|-|-|-|-|-|-|
> ||**Gauss.**|**Shot**|**Impl.**|**Defoc.**|**Glass**|**Motion**|**Zoom**|**Snow**|**Frost**|**Fog**|**Brit.**|**Contr.**|**Elas.**|**Pix.**|**JPEG**|**Avg.**|
> |TED_Random|43.86|46.68|41.82|40.13|49.16|48.10|43.77|40.82|40.34|40.14|63.24|40.10|40.21|63.60|60.95|46.86|
> |TED_(1+1)ES|56.10|56.93|56.79|47.20|35.17|53.49|44.82|62.95|63.59|66.94|78.10|33.11|46.22|67.26|67.23|55.73|
> |TED_ZO|55.06|55.88|52.62|45.53|34.41|52.50|43.96|62.39|59.66|64.82|77.59|21.29|45.46|66.48|66.54|53.61|
> |TED_CMA|58.77|59.66|59.50|49.30|36.08|55.35|46.34|65.21|66.40|67.66|80.21|35.96|47.61|69.55|69.68|57.82|
>
> CMA‑ES is also **hardware‑friendly**: it is fully **gradient‑free** and only needs repeated forward passes, which matches our edge constraints. Practically, it has **mature Python and C/C++ implementations**, making it straightforward to integrate both in our research code and in low‑level runtimes for deployment. Thus, CMA‑ES is used as a **practical and stable tool** that fits our TTA formulation best, rather than as an algorithmic novelty.
>
> **[W4] Limited gains for some corruptions and failure modes**
>
> We would like to clarify that, in Table 1, **TED improves performance on all corruption types** on ImageNet‑C: the per‑corruption gains range from **+1.30% to +3.92%**. Thus there are no cases with negligible or negative improvement in this benchmark.
>
> What we intended to discuss in Sec. 4.3 are a few **extremely hard cases where the baseline accuracy is already near random**, such as **Glass Blur on MobileNet‑V4 (3.03%)** or **Contrast on ResNet‑50 (0.72%)**. In these settings, the feature representations are already **heavily degraded**, so there is very little meaningful structure left for any TTA method to exploit. Consistent with this, TED brings **little to no** improvement on these particular corruptions, and the accuracy remains close to the original baseline. This indicates that, under such extreme degradations, improving the **backbone robustness** is likely more critical than test‑time adaptation.

---

> ### Author Response · Authors · 2025-11-20
> **Thank you; Address your concerns [3/4]**
>
> **[W5] Dataset diversity**
>
> Besides ImageNet‑C, we already evaluate on **ImageNet‑V2**, **ImageNet‑R**, and **ImageNet‑Sketch**, which introduce natural re‑sampling, style/rendering, and sketch‑like domain shifts, as well as on a speech keyword spotting task. These results indicate that TED is **not tied to a specific dataset or modality**.
>
> Regarding CIFAR10‑C / CIFAR100‑C, they use the same corruption types as ImageNet‑C but on smaller 10/100‑class datasets; ImageNet‑C with 1,000 classes is strictly more challenging in this family. Following the reviewer’s suggestion for harder domain‑shift benchmarks, we instead add experiments on DomainNet‑126. Across four source domains, TED is competitive in the batch setting, and in the single‑instance setting it effectively prevents catastrophic forgetting, while baselines degrade severely. These results further support the generality of our conclusions.
>
> **T2. Performance Comparison on DomainNet-126 (Painting as source)**
> |Method|**Real**|**Sketch**|**Clipart**|**Avg.**|
> |-|-|-|-|-|
> |No&nbsp;Adapt|74.84|49.70|53.26|59.27
> |SAR (Bs=64)|73.58|53.96|53.50|60.35
> |ROID (Bs=64)|75.04|57.30|57.24|63.19
> |SAR (Bs=1)|6.42|4.24|3.22|4.63
> |ROID (Bs=1)|0.48|0.16|0.16|0.27
> |MEMO|40.38|17.82|25.42|27.87
> |TED (Bs=64)|78.24|53.74|55.50|62.49
> |TED (Bs=1)|76.08|52.94|54.22|61.08
>
> **T3. Performance Comparison on DomainNet-126 (Clipart as source)**
> |Method|**Sketch**|**Real**|**Paiting**|**Avg.**|
> |-|-|-|-|-|
> |No&nbsp;Adapt|46.16|60.50|43.84|50.17
> |SAR (Bs=64)|48.80|64.36|47.34|53.50
> |ROID (Bs=64)|51.24|64.64|49.14|55.01
> |SAR (Bs=1)|4.98|5.80|4.38|5.05
> |ROID (Bs=1)|0.16|0.48|0.30|0.31
> |MEMO|40.38|15.48|27.92|12.92|18.77
> |TED (Bs=64)|47.24|63.24|46.08|52.19
> |TED (Bs=1)|47.20|62.44|44.74|51.46
>
> **T4. Performance Comparison on DomainNet-126 (Real as source)**
> |Method|**Clipart**|**Paiting**|**Sketch**|**Avg.**|
> |-|-|-|-|-|
> |No&nbsp;Adapt|55.62|61.88|47.84|55.11
> |SAR (Bs=64)|54.36|62.90|48.40|55.22
> |ROID (Bs=64)|57.48|65.18|52.66|58.44
> |SAR (Bs=1)|3.74|4.66|3.58|3.99
> |ROID (Bs=1)|0.16|0.30|0.16|0.21
> |MEMO|40.38|26.12|22.92|15.86|21.63
> |TED (Bs=64)|56.40|63.28|49.82|56.50
> |TED (Bs=1)|56.50|63.12|49.16|56.26
>
> **T5. Performance Comparison on DomainNet-126 (Sketch as source)**
> |Method|**Clipart**|**Paiting**|**Sketch**|**Avg.**|
> |-|-|-|-|-|
> |No&nbsp;Adapt|48.48|54.88|59.32|54.23
> |SAR (Bs=64)|57.56|58.98|67.92|61.49
> |ROID (Bs=64)|58.76|61.44|68.58|62.93
> |SAR (Bs=1)|4.78|4.50|5.26|4.85
> |ROID (Bs=1)|0.30|0.16|0.48|0.31
> |MEMO|13.92|26.00|21.68|20.53
> |TED (Bs=64)|56.52|61.32|64.46|60.77
> |TED (Bs=1)|54.44|59.50|63.50|59.15

---

> ### Author Response · Authors · 2025-11-20
> **Thank you; Address your concerns [4/4]**
>
> **[W6] Comparison with additional baselines**
>
> We appreciate the suggestion and have added explicit comparisons with [4] and [5]. Since both methods are designed for **continual TTA**, we evaluate them in the **continual single‑instance** setting and the **continual batch** setting, following the reviewer 8Tmh’s comments.
>
> In the continual single‑instance regime, both [4] and [5] suffer from **severe catastrophic forgetting** and quickly collapse, effectively failing to adapt, while TED remains stable because it only updates a tiny latent state and keeps the backbone frozen. In the continual batch setting, TED’s accuracy is **comparable to or between** [4] and [5], but it uses **≈15× less runtime memory**, making it significantly more suitable for resource‑constrained deployment.
>
> **T6. Continual single-instance setting. Evaluated on ImageNet-C with ViT-Base.**
> |Method|Memory||Noise|||Blur||||Weather||||Digital|||Acc.
> |-|-|-|-|-|-|-|-|-|-|-|-|-|-|-|-|-|-|
> |||**Gauss.**|**Shot**|**Impl.**|**Defoc.**|**Glass**|**Motion**|**Zoom**|**Snow**|**Frost**|**Fog**|**Brit.**|**Contr.**|**Elas.**|**Pix.**|**JPEG**|**Avg.**|
> |No&nbsp;Adapt||55.34|56.23|56.01|46.48|34.78|52.87|44.20|62.39|62.66|65.56|77.70|32.04|45.73|66.72|66.67|55.03|
> |ROID|1345MB|0.10|0.08|0.08|0.08|0.08|0.08|0.08|0.08|0.08|0.08|0.08|0.08|0.08|0.08|0.08|0.08|
> |CVPR'24|4033MB|3.86|6.68|1.82|0.13|9.16|8.10|3.77|0.82|0.34|0.14|79.24|0.10|0.21|73.60|70.95|17.26|
> |FOA|702MB|6.94|1.74|1.42|1.72|0.40|0.48|0.68|0.86|0.72|0.98|2.04|0.88|0.86|0.22|0.96|1.39|
> |ZOA|846MB|0.06|0.06|0.06|0.06|0.06|0.06|0.06|0.06|0.06|0.06|0.06|0.06|0.06|0.06|0.06|0.06|
> |T3A|1128MB|55.18|56.72|56.00|38.88|32.96|50.96|42.82|60.14|60.18|64.22|76.48|40.24|43.12|66.60|68.48|54.20|
> |SAR|1288MB|59.08|60.52|59.36|45.52|57.26|58.56|57.12|62.74|66.66|68.68|78.78|6.68|67.16|72.40|71.56|59.47|
> |TED|696MB|61.78|62.40|62.98|51.82|39.50|57.74|47.44|68.92|68.52|68.72|80.80|31.74|56.20|70.30|67.94|59.79|
>
> **T7. Continual batch setting. Evaluated on ImageNet-C with ViT-Base.**
> |Method|Memory||Noise|||Blur||||Weather||||Digital|||Acc.
> |-|-|-|-|-|-|-|-|-|-|-|-|-|-|-|-|-|-|
> |||**Gauss.**|**Shot**|**Impl.**|**Defoc.**|**Glass**|**Motion**|**Zoom**|**Snow**|**Frost**|**Fog**|**Brit.**|**Contr.**|**Elas.**|**Pix.**|**JPEG**|**Avg.**|
> |No&nbsp;Adapt||55.34|56.23|56.01|46.48|34.78|52.87|44.20|62.39|62.66|65.56|77.70|32.04|45.73|66.72|66.67|55.03|
> |ROID|10866MB|60.68|63.04|63.00|56.60|55.84|62.72|58.86|66.78|66.70|70.76|79.24|12.14|57.02|69.46|71.04|60.93|
> |CVPR'24|19489MB|58.52|63.92|64.46|50.34|40.66|56.58|44.94|67.62|68.42|66.48|77.74|39.54|51.66|68.46|69.70|59.27|
> |FOA|1300MB|57.14|61.62|61.96|51.98|42.78|57.48|54.08|65.98|68.44|65.60|79.48|62.78|52.20|70.32|72.20|61.60|
> |ZOA|1734MB|58.00|58.56|59.38|47.98|39.10|56.38|49.38|65.68|63.48|62.68|79.34|49.02|50.76|66.84|70.34|58.46|
> |T3A|6818MB|58.70|61.18|61.40|47.48|48.88|58.38|50.92|62.82|62.30|64.90|78.32|60.40|54.78|67.50|69.44|60.49|
> |SAR|1458MB|55.18|56.72|56.02|38.76|33.04|50.98|42.72|60.12|60.12|64.22|76.48|40.22|43.10|66.62|68.44|54.18|
> |TED|1290MB|61.38|61.98|62.64|51.08|39.16|57.70|47.20|68.76|68.60|71.50|79.96|30.78|55.78|69.28|67.08|59.53|
>
> Reference:\
> [1] On calibration of modern neural networks, ICML 2017.\
> [2] A simple unified framework for detecting out-of-distribution samples and adversarial attacks, NeurIPS 2018.\
> [3] Prevalence of neural collapse during the terminal phase of deep learning training, NAS 2020.\
> [4] A Versatile Framework for Continual Test-Time Domain Adaptation: Balancing Discriminability and Generalizability, CVPR 2024\
> [5] Universal Test-time Adaptation through Weight Ensembling, Diversity Weighting, and Prior Correction, WACV 2024\

---

> ### Comment · Reviewer_QpSX · 2025-11-26
>
> I have read the rebuttal coments, I keep my score interms of the novelty and experimental results.

---

> > ### Author Response · Authors · 2025-11-28
> > **Thank you and further response**
> >
> > Dear Reviewer,
> >
> > We thank the reviewer for reading our rebuttal. Given that we have fully addressed your specific requests from the initial review, we respectfully seek further clarification to ensure we haven't misunderstood your remaining concerns. We believe these new results strongly support the validity of our method.
> >
> > ---
> >
> > **1. Regarding "Experimental Results"**
> >
> > In your initial review, you raised concerns about dataset difficulty and missing baselines. In response, we provided:
> >
> > - **More Challenging Benchmarks:** As requested, we evaluated on DomainNet-126 (Tables T2-T5 in the rebuttal), which is widely recognized as a harder domain-shift benchmark than standard corruption sets. TED significantly outperforms the baselines (e.g., preventing the collapse seen in SAR/ROID) in the single-instance setting.
> >
> > - **New SOTA Comparisons:** We implemented and compared with CVPR'24 [1] and WACV'24 [2] (Tables T6-T7). The results show that these methods suffer from severe catastrophic forgetting in the single-instance setting, whereas TED remains stable and accurate.
> >
> > Could you please specify which part of these new experimental results is considered insufficient? We are committed to addressing any specific technical gaps you identify.
> >
> > ---
> >
> > **2. Regarding "Novelty": A Paradigm Shift for Edge Deployment**
> >
> > We respectfully reiterate that our novelty should be viewed through the lens of **edge constraints** rather than component complexity.
> >
> > While PCA and CMA-ES are existing tools, applying them to achieve **gradient-free, memory-constant adaptation** that rivals heavy gradient-based methods is a **non-trivial system-level** contribution.
> >
> > Most existing TTA methods (including the ones you suggested) fail in strict edge environments (due to memory overhead or backpropagation requirements). TED is novel because it effectively solves this **"Deployment Gap"**, offering a capability that previous methods simply do not have.
> >
> > ---
> >
> > We sincerely hope these clarifications highlight the distinct value of our work. If there are specific failure modes or metrics in our new data that concern you, we would be grateful for the opportunity to discuss them.
> >
> > Sincerely,
> >
> > The Authors
> >
> > Reference:\
> > [1] A Versatile Framework for Continual Test-Time Domain Adaptation: Balancing Discriminability and Generalizability, CVPR 2024\
> > [2] Universal Test-time Adaptation through Weight Ensembling, Diversity Weighting, and Prior Correction, WACV 2024

---

### Official Review · Reviewer_qiYF · 2025-10-31

**Soundness:** 1
**Presentation:** 2
**Contribution:** 1
**Rating:** 2
**Confidence:** 4

**Summary:**

This paper targets single-instance test-time adaptation (TTA) on edge devices. It leverages source-domain latent statistics computed offline, and at test time keeps model parameters frozen while forward-only optimizing a low-dimensional coordinate to align the target latent toward the source distribution. The authors report accuracy/efficiency gains over prior TTA methods on image classification and keyword spotting, and support practicality with on-device experiments using the quantized variant (QTED).

**Strengths:**

- The method is simple, facilitating implementation and portability, and achieves computational efficiency by updating only the k-dimensional adaptation variable at test time rather than the model parameters.

- The approach is architecture-agnostic, applying to diverse backbones (e.g., CNNs, Transformers, RNN/LSTM encoders) as long as a latent representation is available.

**Weaknesses:**

- Practicality limited by dependence on source statistics. The paper’s central contribution hinges on source-domain latent statistics. Appendix C.3 shows clear sensitivity to the number of source samples used to compute those statistics (e.g., 5k is insufficient, and Table 1 relies on a full 50k). Moreover, obtaining the statistics requires forwarding the entire source dataset through the encoder to build the latent mean and subspace, which is often infeasible in real deployments where source data cannot be accessed or processed post-training. Without these statistics, the method cannot run, which substantially undermines practical applicability.

- The core contribution is test-time latent-space modification guided by source statistics, which is not new, and the optimization relies on an existing algorithm (CMA-ES). Moreover, the on-device deployment appears to hinge primarily on reducing parameter bit-width (quantization). Overall, the contribution reads as incremental rather than a substantive advance.

- Potentially unfair comparisons. Many baselines (e.g., T3A/CoTTA/SAR/MEMO/FOA) do not use source data or source statistics, whereas the proposed method starts with source statistics. Comparing across these different information regimes risks unfair conclusions. It would also help to compare against work that introduces single-instance backpropagation-free TTA via normalization and to explain the differences from [r1].

- Quantization fairness and presentation. Appendix A discusses quantization (QTED), yet the main text does not foreground that quantization is required for the Zynq-7020 deployment, which can make it appear that the method is edge-deployable without quantization. Since quantization is not the paper’s contribution, claiming on-device evidence without evaluating baselines under matched quantization/precision and identical budgets is unfair.

References:
[r1] Mirza, M. J., Micorek, J., Possegger, H., & Bischof, H. (2022). The Norm Must Go On: Dynamic Unsupervised Domain Adaptation by Normalization. In Proceedings of the IEEE/CVF Conference on Computer Vision and Pattern Recognition (CVPR 2022).

**Questions:**

Q1. To mitigate concerns about reliance on source statistics, can you provide sensitivity analyses over the number of source samples N and the subspace dimension k, and also justify the use of source statistics in your problem setting with a clear rationale and deployment assumptions?

Q2. On quantization, can you show matched-precision comparisons where all baselines are quantized and deployed on the same edge device with the same bit-width and compute/memory budgets as QTED?

Q3. On technical novelty, can you explicitly state what is novel beyond latent-space modification with source statistics and the use of CMA-ES, and provide supporting analysis or evidence?

---

> ### Author Response · Authors · 2025-11-20
> **Thank you; Address your concerns [1/4]**
>
> Dear Reviewer,
>
> Thank you for your thoughtful comments, which have helped us make our work clearer and more solid. We have carefully addressed all your concerns in detail. In the revised manuscript, changes are highlighted in green and yellow, with yellow indicating changes also requested by other reviewers.
>
> **[W1&W3&Q1] Source statistics and its sensitivity.**
>
> We would like to clarify that TED **does not require any source-domain samples at test time**. Source data are used only once **offline** to estimate the latent mean and principal subspace, exactly analogous to how many methods (including FOA) precompute CLS statistics or BN running statistics on the training server. This estimation can be done **during training** (by logging latent features or their running moments) or immediately after, without any additional access to source data on the deployed device. For ViT‑Base, the stored subspace is about **0.05 MB vs. 330.28 MB** for the model (**≈0.01% extra**), so both storage and loading on edge devices are negligible, and the number of source samples used offline does **not** change the test‑time memory/compute of TED (which depends only on *k* and latent dimension *D*).
>
> Regarding sensitivity to the number of source samples *N*, our original Appendix C.3 only reported results down to 5k samples and (incorrectly) gave the impression that “smaller *N* ⇒ worse performance”. Motivated by your and the reviewer 8Tmh’s comment, we ran additional experiments with **N = 20, 30, 50, 100, 500** (each repeated with three random seeds), and observed a **non‑monotonic** behavior: very small *N* (e.g., 20–100) already achieve accuracy comparable to, or even **better than**, using the full 50k set, while around **1k-5k** happens to be a local minimum. This pattern is consistent across seeds and between loss and accuracy, so it is **not a random artifact** or pure **overfitting** to entropy.
>
> A plausible explanation is that with very small *N*, the subspace captures **only the dominant, class‑discriminative directions**, which are especially helpful for entropy‑guided adaptation; as *N* grows, additional, less stable directions can temporarily hurt, before being averaged out again when *N* becomes very large. The detailed Table T2 supports this interpretation quantitatively. The performance is highest when using only 20–50 source samples (≈59.14%), drops markedly in the mid‑N regime (down to 51.5% at N=1k), and then partially recovers when using all 50k samples (57.82%), but still does not surpass the small‑N setting. The Fog corruption shows this non‑monotonic behavior most clearly: accuracy remains high and relatively stable for small *N* (72.8/72.6/71.7 at *N*=20/30/50), but collapses to 16.8/7.3/29.4 at *N*=500/1k/3k, exactly in the regime where the subspace first gains many additional, less stable directions that are not reliably aligned with class‑discriminative structure and can easily mislead the entropy‑only objective. When *N* further increases to 50k, these extra directions are statistically “averaged out”, the subspace becomes stable again, and Fog accuracy recovers to 67.7. Overall, this “good–bad–recovering” pattern, together with the robustness of performance for very small *N* and for large *N*, is exactly what we would expect if mid‑*N* subspaces introduce unstable directions that misguide the entropy‑only adaptation, whereas very small *N* and very large *N* correspond to, respectively, a highly constrained and a fully averaged‑out (and thus stable, relatively insensitive) subspace.
>
> To further probe this effect and the reviewer’s question about *k*-sensitivity, Table T3 varies the subspace dimension *k* at small (N=30), medium (N=1k), and larger (N=5k) sample sizes. Consistent with the above picture, accuracy is almost flat in *k* for very small *N*, but for *N*=1k performance drops sharply as *k* increases (additional unstable directions), whereas for *N*=5k the dependence on *k* becomes monotone and much milder, indicating that once the subspace is sufficiently averaged, enlarging *k* no longer hurts adaptation.
>
> We will update Appendix C.3 with these new results and a brief discussion, and emphasize in the main text that **only tens of source samples** is sufficient in practice, and that this is an **offline modeling choice** rather than a test‑time requirement.
>
> Due to the characters limitation, the three tables mentioned above is in the next reply. Thanks for understanding.

---

> ### Author Response · Authors · 2025-11-20
> **Thank you; Address your concerns [2/4]**
>
> **T1. Performance comparison on ImageNet-C with ViT-Base model using different *N***
> |*N*|50k|40k|30k|20k|10k|5k|3k|1k|500|100|50|30|20|
> |-|-|-|-|-|-|-|-|-|-|-|-|-|-|
> |Acc.|57.82|57.69|57.28|56.89|56.53|55.94|53.41|51.52|54.75|58.66|59.08|**59.14**|59.13|
> |Avg. Loss|2.33|2.45|2.52|2.59|2.43|2.62|2.77|2.75|2.38|2.20|2.23|2.25|2.19|
>
> **T2. Details of Performance comparison on ImageNet-C with ViT-Base model using different *N***
> |*N*|Noise|||Blur||||Weather||||Digital||||Acc.|
> |-|-|-|-|-|-|-|-|-|-|-|-|-|-|-|-|-|
> ||**Gauss.**|**Shot**|**Impl.**|**Defoc.**|**Glass**|**Motion**|**Zoom**|**Snow**|**Frost**|**Fog**|**Brit.**|**Contr.**|**Elas.**|**Pix.**|**JPEG**|**Avg.**|
> |20|60.69|60.50|61.24|50.07|37.76|56.36|47.39|65.71|67.23|72.79|81.07|35.76|48.54|70.50|71.36|59.13|
> |30|60.69|60.41|61.24|50.27|37.80|56.39|47.40|65.61|67.05|72.61|81.01|36.37|48.44|70.53|71.29|59.14|
> |50|60.83|60.53|61.39|50.33|37.73|56.50|47.39|65.89|67.07|71.65|80.81|35.66|48.60|70.47|71.31|59.08|
> |100|60.65|60.59|61.28|50.06|37.57|56.32|47.39|65.30|66.96|68.88|81.00|33.64|48.50|70.40|71.30|58.66|
> |500|60.30|59.67|60.66|49.87|37.16|56.06|47.14|65.27|66.03|16.79|80.56|33.17|47.87|70.03|70.71|54.75|
> |1000|59.22|59.11|59.82|48.99|37.19|55.46|46.76|64.73|65.31|7.26|80.21|1.02|48.10|69.28|70.31|51.52|
> |3000|59.03|59.11|59.68|49.24|37.19|55.44|46.85|64.66|65.04|29.36|79.79|8.32|47.94|69.22|70.23|53.41|
> |50k|58.77|59.66|59.50|49.30|36.08|55.35|46.34|65.21|66.40|67.66|80.21|35.96|47.61|69.55|69.68|57.82|
>
> **T3. Accuracy on ImageNet‑C vs. source sample count *N* and subspace dimension *k* (ViT‑Base)**
> |*N* \ *k*|4|8|16|32|
> |-|-|-|-|-|
> |30|58.03|59.09|59.14|59.15|
> |1k|54.63|56.25|51.52|48.74|
> |5k|55.39|56.61|57.82|57.90|

---

> ### Author Response · Authors · 2025-11-20
> **Thank you; Address your concerns [3/4]**
>
> **[W2&Q3] Technical novelty.**
>
> We agree that PCA and CMA‑ES are standard techniques, and we do not claim novelty on these tools themselves. The novelty of our work lies in a **non‑trivial TTA paradigm**: under strict edge constraints, we show that it is sufficient to adapt only **a handful of latent scalars** per sample ($16$ parameters for ViT, $2$ for LSTM) while keeping the entire backbone frozen.
>
> This leads to properties that standard “PCA + optimizer” combinations do not offer:
> 1. **Extreme parameter and memory efficiency.** The adaptable state is a fixed‑size latent vector whose dimension is independent of the backbone size, which makes the method highly hardware‑friendly and easy to deploy on constrained devices.
> 2. **Stability even under single‑instance or continual updates.** Because we only update a tiny latent vector and never touch backbone weights, our method naturally avoids catastrophic forgetting and instability that typically arise when repeatedly adapting large parameter sets or normalization layers in edge scenarios.
>
> Despite this **ultra‑low‑dimensional update**, the framework consistently provides strong TTA across **multiple architectures** (rather than being limited to models with BN layers or prompt‑based designs) and **application domains**, suggesting that the formulation itself is non‑trivial rather than a superficial combination of existing components. We will clarify in the revision that our contribution is this **edge‑oriented, minimal‑state latent adaptation paradigm**, not new PCA or optimization algorithms.
>
> **[W2&W4&Q2] TED's Edge Deployability.**
>
> Our core contribution is algorithmic: TED performs test‑time adaptation by updating only a k‑dimensional latent coordinate vector via forward passes, with no backpropagation and no weight updates. This design already yields large reductions in GFLOPs, peak memory, and runtime in the **standard FP32 GPU setting** (Table 4 and Fig. 1), completely **independent** of any quantization.
>
> The ZYNQ‑7020 experiments are intended as a proof‑of‑concept demonstration that this forward‑only, few‑parameter update can run on a fixed‑point, low‑precision SoC, not as the primary source of our efficiency gains. **It does not rely on QTED**. On ZYNQ, we simply use the native fixed‑point precision supported by the platform, i.e., the computation is automatically truncated by the hardware’s arithmetic units; we do not apply any additional, dedicated model‑compression or network‑quantization scheme beyond this. This is also why the gains on ZYNQ are smaller than on GPU: the change in numeric precision hurts both the baseline and TED.
>
> Accordingly, we treat the on‑device results as a deployment case study rather than a full device‑level benchmark of all TTA methods. Most existing TTA baselines require backpropagation and large activation buffers, which our hard-coded edge platforms cannot support. This deployability gap is exactly the motivation of our work: TED is designed for TTA in such constrained settings. We therefore compare to gradient-based methods in the standard GPU setting, and use on-device experiments only to demonstrate that TED is practical under realistic edge constraints.
>
> **[W2&W4] On the role of QTED as a hardware‑oriented extension.**
>
> QTED is **a separate, forward‑looking design for hardware co‑design**, not a prerequisite for our main results. Appendix A introduces QTED‑V1 and QTED‑V2 as additional variations that explore how far we can push hardware efficiency if we co‑design the algorithm and the accelerator. In QTED‑V1, the optimization variable $p$ is quantized to 1 bit per coordinate, effectively turning the k‑dimensional vector into k binary switches. This is meant to highlight that a future, dedicated accelerator could implement the adaptation logic using extremely simple hardware primitives, not that current ZYNQ experiments already depend on this.

---

> ### Author Response · Authors · 2025-11-20
> **Thank you; Address your concerns [4/4]**
>
> **[W3] Concerns on baselines.**
>
> We believe the comparison is fair once the actual information and cost are made explicit.
>
> 1. Baselines are also “source‑dependent”.
> FOA uses source‑domain statistics for its feature alignment; CoTTA requires a separately trained student/teacher; T3A relies on the source‑trained classifier head as prototypes; many other TTA methods depend on BN running means/variances, which are themselves source statistics stored in the checkpoint. In realistic deployment, all methods start from a source‑trained model with its internal statistics.
>
> 2. Our source statistics are offline, tiny, and test‑time–free.
> TED **never accesses source images or labels at test time**. We only precompute once, offline, **the source latent**, then keep them fixed. These add ≈0.01% to the model size—comparable to BN statistics—and incur no extra source‑data access during adaptation; at test time we only optimize a **k‑dimensional vector**.
>
> 3. Fairness in test‑time regime.
> Under this regime, TED uses the **same kind of source‑trained checkpoint information** as common TTA methods, but adapts far fewer parameters (k scalars, no backprop) than CoTTA/SAR/MEMO/FOA.
>
> Besides, we consider the work DUA[r1]. DUA updates **BN statistics** per test sample and is therefore restricted to **BN‑based CNNs**, and cannot be applied to backbones ViT‑Base (LayerNorm) or LSTM (no normalization layers). By contrast, TED operates purely in the **latent feature space** and only assumes an encoder–decoder interface, making it applicable to **CNNs, ViTs, and RNN/LSTM architectures**. Moreover, [r1] attains robustness by building a **large augmentation batch per test point** (e.g., 64 views for ImageNet series), requiring extra 63 forward passes and repeated normalization updates, which significantly increases latency and memory; TED instead optimizes a **k‑dimensional latent vector with a small number of forward passes**, yielding a more resource‑friendly single‑instance, backprop‑free TTA scheme suitable for edge deployment.
>
> **T4. Performance comparison with DUA on ImageNet-C.**
> |Method|Memory|Latency|Acc.|
> |-|-|-|-|
> |DUA|3420MB|0.703s|56.46%|
> |TED|696MB|0.042s|57.82%|

---

> > ### Author Response · Authors · 2025-11-28
> > **Follow-up on our rebuttal**
> >
> > Dear Reviewer,
> >
> > Thank you again for your time and constructive feedback during the initial review.
> >
> > We are writing to gently follow up and ensure that our rebuttal and the revised manuscript have successfully addressed your concerns. We have provided detailed responses to your questions, particularly clarifying the minimal dependence on source statistics (new experiments showing efficacy with only N=20 offline samples) and the independence of our method from quantization for deployment.
> >
> > We value your feedback and remain fully available to answer any further questions or run additional checks before the discussion period closes.
> >
> > Best regards,
> >
> > The Authors

---

> > > ### Comment · Reviewer_qiYF · 2025-11-28
> > > **Thanks for the rebuttal**
> > >
> > > Some of my concerns are addressed with your sincere rebuttal, but I would keep my score because of the limited technical novelty and remaining concerns about the weakened practical justification for relying on offline source statistics and the fact that the N-sensitivity analysis is shown only on ImageNet-C, which makes the claimed behavior look potentially dataset-specific.

---

### Official Review · Reviewer_8Tmh · 2025-10-31

**Soundness:** 2
**Presentation:** 2
**Contribution:** 2
**Rating:** 4
**Confidence:** 4

**Summary:**

This paper proposes TED (Test-time adaptation for Edge Devices), a gradient-free single-instance test-time adaptation method. TED performs adaptation by correcting latent feature coordinates within a principal component subspace computed from the source domain. The method uses CMA-ES optimization to minimize prediction entropy without requiring backpropagation, aiming to achieve efficient and lightweight adaptation suitable for edge devices. Experiments are conducted on several ImageNet variants and keyword spotting tasks under a single-image test-time adaptation setting, showing moderate accuracy gains compared to other gradient-free baselines.

**Strengths:**

1. The paper provides a clear and intuitive formulation of latent-space adaptation based on principal component correction.
2. The proposed method is simple and broadly applicable, requiring no architectural modifications and minimal additional computation.
3. The integration of PCA-based latent subspace modeling with a backpropagation-free optimization framework is methodologically sound and represents a reasonable design choice for efficient test-time adaptation.

**Weaknesses:**

1. I have concerns regarding the single-image setting adopted in this paper. In realistic deployment scenarios, it is highly unlikely that a model would only ever process one test sample at a time without any opportunity for continuous adaptation. Even under the most restrictive conditions, a model can accumulate several steps before resetting, which would allow existing methods such as SAR[1] to significantly improve their stability and learning efficiency. The authors should further justify the motivation and practical relevance of this assumption.
2. Compared with the strict single-image setting, adopting a batch size = 1 constraint during continuous adaptation, together with label shift scenarios, would more closely reflect the characteristics of edge devices and real-world streaming environments. The paper should include corresponding experiments under these more realistic settings to evaluate TED’s robustness and effectiveness in continual adaptation.
3. In contrast to FOA[2], which only requires 32 source-domain samples to compute the necessary statistics, TED needs up to 5000 source-domain samples to reach 55.94% accuracy (compared to 55.03% for No Adapt). Accessing source-domain data at such a scale is prohibitively expensive and nearly unacceptable for practical applications of test-time adaptation.
4. The paper lacks comparisons with several single-instance normalization calibration or lightweight TTA methods, such as IABN[A], TTN[B], and ZOA[C]. Including these baselines would provide a more comprehensive evaluation and clarify the relative advantages of TED in limited-data or resource-constrained scenarios.
5. The experiments claiming the effectiveness of TED across diverse network architectures and edge devices only compare against the No Adapt baseline, without including other representative TTA methods.
6. Both the main text and the appendix omit important hyperparameter and tuning details for all baseline methods, including learning rates, update steps, and regularization weights. The absence of this information severely limits reproducibility and transparency. A complete configuration table should be provided in the supplementary material.

[1] Towards stable test-time adaptation in dynamic wild world. ICLR 2023.

[2] Test-Time Model Adaptation with Only Forward Passes. ICML 2024.

[A] NOTE: Robust Continual Test-time Adaptation  Against Temporal Correlation. NIPS 2022.

[B] TTN: A Domain-Shift Aware  Batch Normalization in Test-Time Adaptation. ICLR 2023.

[C] Test-Time Model Adaptation for Quantized Neural Networks. ACMMM 2025.

**Questions:**

1. In all reported settings, TED resets the model immediately after adapting to a single test sample. I would like to know whether TED itself possesses the capability for continuous or streaming adaptation.
2. In the ablation study, some larger combinations of (e.g. k8n10) show a noticeable drop in performance (50.01%). Intuitively, increasing k or n within a reasonable range should improve performance—why does it instead lead to a significant degradation in this case?

---

> ### Author Response · Authors · 2025-11-20
> **Thank you; Address your concerns [1/4]**
>
> Dear Reviewer,
>
> Thank you for your thoughtful comments, which have helped us make our work clearer and more solid. We have carefully addressed all your concerns in detail. In the revised manuscript, changes are highlighted in blue and yellow, with yellow indicating changes also requested by other reviewers.
>
> **[W1&W2&Q1] Motivation and practicality of the strict single-instance setting, and TED’s capability for continual adaptation.**
>
> We focus on a strict single-instance setting because it reflects a realistic and conservative edge-deployment scenario. Our choice is motivated by two aspects:
> 1. **Resource constraints**: On-device memory and energy budgets are tight, so maintaining dynamic feature buffers, EMA models, or large adaptation statistics across samples often competes with model and activation storage.
> 2. **Application patterns**: In many edge image-classification applications (e.g., inspection snapshots, access-control photos) and keyword-spotting systems operating on short audio windows, inference is triggered on small, self-contained units and the device may enter low-power mode between calls, which makes long-horizon continual adaptation impractical.
>
> In addition, this single-instance regime is also well studied on the academic side: several recent works explicitly focus on **single-instance TTA** under similar assumptions [1,2,3], indicating that single-instance adaptation is not only practically motivated but also an established and relevant research setting.
>
> In response to continual adaptation setting, we additionally evaluate TED under continual single-sample and continual batch settings (batch size = 64) without resetting the model between test samples or domains.
> 1. **Continual single-sample**: In this setting several gradient-based TTA methods suffer from catastrophic forgetting when repeatedly updating model parameters on a long test stream. In contrast, TED only optimizes an extremely low-dimensional latent coordinate for each sample while keeping the backbone fixed, which effectively prevents forgetting. As a result, TED achieves state-of-the-art accuracy while using the smallest peak memory among all compared methods.
> 2. **Continual batch**: TED is designed for per-sample coordinate correction, so for a batch we apply a shared shift vector *p* to all samples in the batch. This global shift is less precise than per-sample optimization, yet TED remains highly competitive in terms of accuracy and still exhibits the lowest peak memory footprint in this streaming batch setting.
>
> **T1. Continual single-instance setting. Evaluated on ImageNet-C with ViT-Base.**
> |Method|Memory||Noise|||Blur||||Weather||||Digital|||Acc.
> |-|-|-|-|-|-|-|-|-|-|-|-|-|-|-|-|-|-|
> |||**Gauss.**|**Shot**|**Impl.**|**Defoc.**|**Glass**|**Motion**|**Zoom**|**Snow**|**Frost**|**Fog**|**Brit.**|**Contr.**|**Elas.**|**Pix.**|**JPEG**|**Avg.**|
> |No&nbsp;Adapt||55.34|56.23|56.01|46.48|34.78|52.87|44.20|62.39|62.66|65.56|77.70|32.04|45.73|66.72|66.67|55.03|
> |ROID|1345MB|0.10|0.08|0.08|0.08|0.08|0.08|0.08|0.08|0.08|0.08|0.08|0.08|0.08|0.08|0.08|0.08|
> |CVPR'24|4033MB|3.86|6.68|1.82|0.13|9.16|8.10|3.77|0.82|0.34|0.14|79.24|0.10|0.21|73.60|70.95|17.26|
> |FOA|702MB|6.94|1.74|1.42|1.72|0.40|0.48|0.68|0.86|0.72|0.98|2.04|0.88|0.86|0.22|0.96|1.39|
> |ZOA|846MB|0.06|0.06|0.06|0.06|0.06|0.06|0.06|0.06|0.06|0.06|0.06|0.06|0.06|0.06|0.06|0.06|
> |T3A|1128MB|55.18|56.72|56.00|38.88|32.96|50.96|42.82|60.14|60.18|64.22|76.48|40.24|43.12|66.60|68.48|54.20|
> |SAR|1288MB|59.08|60.52|59.36|45.52|57.26|58.56|57.12|62.74|66.66|68.68|78.78|6.68|67.16|72.40|71.56|59.47|
> |TED|696MB|61.78|62.40|62.98|51.82|39.50|57.74|47.44|68.92|68.52|68.72|80.80|31.74|56.20|70.30|67.94|59.79|
>
> **T2. Continual batch setting. Evaluated on ImageNet-C with ViT-Base.**
> |Method|Memory||Noise|||Blur||||Weather||||Digital|||Acc.
> |-|-|-|-|-|-|-|-|-|-|-|-|-|-|-|-|-|-|
> |||**Gauss.**|**Shot**|**Impl.**|**Defoc.**|**Glass**|**Motion**|**Zoom**|**Snow**|**Frost**|**Fog**|**Brit.**|**Contr.**|**Elas.**|**Pix.**|**JPEG**|**Avg.**|
> |No&nbsp;Adapt||55.34|56.23|56.01|46.48|34.78|52.87|44.20|62.39|62.66|65.56|77.70|32.04|45.73|66.72|66.67|55.03|
> |ROID|10866MB|60.68|63.04|63.00|56.60|55.84|62.72|58.86|66.78|66.70|70.76|79.24|12.14|57.02|69.46|71.04|60.93|
> |CVPR'24|19489MB|58.52|63.92|64.46|50.34|40.66|56.58|44.94|67.62|68.42|66.48|77.74|39.54|51.66|68.46|69.70|59.27|
> |FOA|1300MB|57.14|61.62|61.96|51.98|42.78|57.48|54.08|65.98|68.44|65.60|79.48|62.78|52.20|70.32|72.20|61.60|
> |ZOA|1734MB|58.00|58.56|59.38|47.98|39.10|56.38|49.38|65.68|63.48|62.68|79.34|49.02|50.76|66.84|70.34|58.46|
> |T3A|6818MB|58.70|61.18|61.40|47.48|48.88|58.38|50.92|62.82|62.30|64.90|78.32|60.40|54.78|67.50|69.44|60.49|
> |SAR|1458MB|55.18|56.72|56.02|38.76|33.04|50.98|42.72|60.12|60.12|64.22|76.48|40.22|43.10|66.62|68.44|54.18|
> |TED|1290MB|61.38|61.98|62.64|51.08|39.16|57.70|47.20|68.76|68.60|71.50|79.96|30.78|55.78|69.28|67.08|59.53|
>
> Reference:\
> [1] DUA, CVPR 2022. \
> [2] MEMO, NeurIPS 2022 \
> [3] SPACE, NeurIPS 2025

---

> ### Author Response · Authors · 2025-11-20
> **Thank you; Address your concerns [2/4]**
>
> **[W3] Source-domain sample requirement.**
>
> We would like to clarify that TED **does not require any source-domain samples at test time**. Source data are used only once **offline** to estimate the latent mean and principal subspace, exactly analogous to how many methods (including FOA) precompute CLS statistics or BN running statistics on the training server. This estimation can be done **during training** (by logging latent features or their running moments) or immediately after, without any additional access to source data on the deployed device. For ViT‑Base, the stored subspace is about **0.05 MB vs. 330.28 MB** for the model (**≈0.01% extra**), so both storage and loading on edge devices are negligible, and the number of source samples used offline does **not** change the test‑time memory/compute of TED (which depends only on *k* and latent dimension *D*).
>
> Regarding sensitivity to the number of source samples *N*, our original Appendix C.3 only reported results down to 5k samples and (incorrectly) gave the impression that “smaller *N* ⇒ worse performance”. Motivated by your and the reviewer qiYF’s comment, we ran additional experiments with **N = 20, 30, 50, 100, 500** (each repeated with three random seeds), and observed a **non‑monotonic** behavior: very small *N* (e.g., 20–100) already achieve accuracy comparable to, or even **better than**, using the full 50k set, while around **1k-5k** happens to be a local minimum. This pattern is consistent across seeds and between loss and accuracy, so it is **not a random artifact** or pure **overfitting** to entropy.
>
> A plausible explanation is that with very small *N*, the subspace captures **only the dominant, class‑discriminative directions**, which are especially helpful for entropy‑guided adaptation; as *N* grows, additional, less stable directions can temporarily hurt, before being averaged out again when *N* becomes very large. The detailed Table T2 supports this interpretation quantitatively. The performance is highest when using only 20–50 source samples (≈59.14%), drops markedly in the mid‑N regime (down to 51.5% at N=1k), and then partially recovers when using all 50k samples (57.82%), but still does not surpass the small‑N setting. The Fog corruption shows this non‑monotonic behavior most clearly: accuracy remains high and relatively stable for small *N* (72.8/72.6/71.7 at *N*=20/30/50), but collapses to 16.8/7.3/29.4 at *N*=500/1k/3k, exactly in the regime where the subspace first gains many additional, less stable directions that are not reliably aligned with class‑discriminative structure and can easily mislead the entropy‑only objective. When *N* further increases to 50k, these extra directions are statistically “averaged out”, the subspace becomes stable again, and Fog accuracy recovers to 67.7. Overall, this “good–bad–recovering” pattern, together with the robustness of performance for very small *N* and for large *N*, is exactly what we would expect if mid‑*N* subspaces introduce unstable directions that misguide the entropy‑only adaptation, whereas very small *N* and very large *N* correspond to, respectively, a highly constrained and a fully averaged‑out (and thus stable, relatively insensitive) subspace.
>
> To further probe this effect and the reviewer’s question about *k*-sensitivity, Table T3 varies the subspace dimension *k* at small (N=30), medium (N=1k), and larger (N=5k) sample sizes. Consistent with the above picture, accuracy is almost flat in *k* for very small *N*, but for *N*=1k performance drops sharply as *k* increases (additional unstable directions), whereas for *N*=5k the dependence on *k* becomes monotone and much milder, indicating that once the subspace is sufficiently averaged, enlarging *k* no longer hurts adaptation.
>
> We will update Appendix C.3 with these new results and a brief discussion, and emphasize in the main text that **only tens of source samples** is sufficient in practice, and that this is an **offline modeling choice** rather than a test‑time requirement.
>
> Due to the characters limitation, the three tables mentioned above is in the next reply. Thanks for understanding.

---

> ### Author Response · Authors · 2025-11-20
> **Thank you; Address your concerns [3/4]**
>
> **T1. Performance comparison on ImageNet-C with ViT-Base model using different *N***
> |*N*|50k|40k|30k|20k|10k|5k|3k|1k|500|100|50|30|20|
> |-|-|-|-|-|-|-|-|-|-|-|-|-|-|
> |Acc.|57.82|57.69|57.28|56.89|56.53|55.94|53.41|51.52|54.75|58.66|59.08|**59.14**|59.13|
> |Avg. Loss|2.33|2.45|2.52|2.59|2.43|2.62|2.77|2.75|2.38|2.20|2.23|2.25|2.19|
>
> **T2. Details of Performance comparison on ImageNet-C with ViT-Base model using different *N***
> |*N*|Noise|||Blur||||Weather||||Digital||||Acc.|
> |-|-|-|-|-|-|-|-|-|-|-|-|-|-|-|-|-|
> ||**Gauss.**|**Shot**|**Impl.**|**Defoc.**|**Glass**|**Motion**|**Zoom**|**Snow**|**Frost**|**Fog**|**Brit.**|**Contr.**|**Elas.**|**Pix.**|**JPEG**|**Avg.**|
> |20|60.69|60.50|61.24|50.07|37.76|56.36|47.39|65.71|67.23|72.79|81.07|35.76|48.54|70.50|71.36|59.13|
> |30|60.69|60.41|61.24|50.27|37.80|56.39|47.40|65.61|67.05|72.61|81.01|36.37|48.44|70.53|71.29|59.14|
> |50|60.83|60.53|61.39|50.33|37.73|56.50|47.39|65.89|67.07|71.65|80.81|35.66|48.60|70.47|71.31|59.08|
> |100|60.65|60.59|61.28|50.06|37.57|56.32|47.39|65.30|66.96|68.88|81.00|33.64|48.50|70.40|71.30|58.66|
> |500|60.30|59.67|60.66|49.87|37.16|56.06|47.14|65.27|66.03|16.79|80.56|33.17|47.87|70.03|70.71|54.75|
> |1000|59.22|59.11|59.82|48.99|37.19|55.46|46.76|64.73|65.31|7.26|80.21|1.02|48.10|69.28|70.31|51.52|
> |3000|59.03|59.11|59.68|49.24|37.19|55.44|46.85|64.66|65.04|29.36|79.79|8.32|47.94|69.22|70.23|53.41|
> |50k|58.77|59.66|59.50|49.30|36.08|55.35|46.34|65.21|66.40|67.66|80.21|35.96|47.61|69.55|69.68|57.82|
>
> **T3. Accuracy on ImageNet‑C vs. source sample count *N* and subspace dimension *k* (ViT‑Base)**
> |*N* \ *k*|4|8|16|32|
> |-|-|-|-|-|
> |30|58.03|59.09|59.14|59.15|
> |1k|54.63|56.25|51.52|48.74|
> |5k|55.39|56.61|57.82|57.90|

---

> ### Author Response · Authors · 2025-11-20
> **Thank you; Address your concerns [4/4]**
>
> **[W4&W5] Completeness of baselines across architectures and devices.**
>
> IABN and TTN are specifically designed for BN-based CNNs and are not applicable to ViT (LayerNorm) or standard LSTMs. The three CNN architectures already used in our paper (ResNet-50, EfficientNet-B0, MobileNet-V4) all employ BatchNorm, so we added IABN and TTN on these models as you suggested. The ResNet-50 results are shown in this reply, and the EfficientNet-B0/MobileNet-V4 results are included in the revised paper; on all three, TED consistently outperforms IABN and TTN. Unlike these BN-specific methods, TED **does not depend on a particular normalization layer** and can be applied in a unified way to ViT, CNNs, and recurrent models. For ZOA, we follow your suggestion and report its performance in our response to **[W1&W2&Q1]**: it is competitive in batch settings but in the single-sample setting again suffers from catastrophic forgetting, whereas TED’s subspace-coordinate updates remain stable.
>
> **T4. Performance on ImageNet-C with ResNet50.**
> |Method|Noise|||Blur||||Weather||||Digital||||Acc.|
> |-|-|-|-|-|-|-|-|-|-|-|-|-|-|-|-|-|
> ||**Gauss.**|**Shot**|**Impl.**|**Defoc.**|**Glass**|**Motion**|**Zoom**|**Snow**|**Frost**|**Fog**|**Brit.**|**Contr.**|**Elas.**|**Pix.**|**JPEG**|**Avg.**|
> |No&nbsp;Adapt|4.47|4.74|4.06|8.11|5.96|9.59|14.74|6.92|14.47|12.32|45.80|0.72|11.08|18.19|32.54|12.91|
> |IABN|7.22|5.10|5.54|5.46|7.38|11.92|15.98|5.74|12.32|14.14|47.10|0.90|7.10|18.26|18.04|12.15|
> |TTN|1.73|5.13|4.90|4.17|6.69|10.16|14.14|6.13|14.52|14.42|49.20|0.22|12.26|22.03|33.07|13.25|
> |SAR|2.34|4.46|4.24|6.56|6.66|14.04|16.34|4.02|14.68|13.26|48.51|0.35|12.43|20.44|36.46|13.65|
> |TED|4.99|5.18|4.41|11.36|7.32|14.26|18.29|7.00|15.32|15.01|55.32|0.25|13.07|23.29|40.97|15.74|
>
> Our hardware experiments are intended as a proof-of-concept that TED can run on real resource-limited devices, rather than as a full device-level benchmark of all TTA methods. Most existing TTA baselines require backpropagation and large activation buffers, which our hard-coded edge platforms cannot support. This deployability gap is exactly the motivation of our work: TED is designed for TTA in such constrained settings. We therefore compare to gradient-based methods in the standard GPU setting, and use on-device experiments only to demonstrate that TED is practical under realistic edge constraints.
>
> **[W6] Missing hyper-parameter and configuration details**
>
> All baselines in our experiments are reproduced using the official implementations and the hyper-parameters recommended in their original papers or public GitHub repositories. The only modifications we make are (i) the batch size, and (ii) whether and when to reset the model state, which are adjusted to match the specific evaluation setting (e.g., single-instance vs. continual). We have added a detailed description of the hyper-parameters and configurations used for each baseline in the revised paper.
>
> **[Q2] Effect of the hyper-parameters k and n**
>
> The subspace dimension *k* trades expressiveness for noise: if *k* is too small, the subspace cannot capture enough useful variation; if *k* is too large, it starts to include many noisy or task-irrelevant directions, and CMA has to search in a harder, higher-dimensional space, which can hurt accuracy. The parameter *n* controls how strongly we optimize the unsupervised proxy objective; very large *n* causes CMA to overfit this proxy (entropy + statistics discrepancy), so the proxy loss keeps improving while true classification performance can degrade. This explains why extreme combinations such as \(k=8, n=10\) lead to a noticeable drop, whereas moderate \(k,n\) lie in a stable “sweet spot”.

---

> > ### Comment · Reviewer_8Tmh · 2025-11-27
> >
> > Thank you for the authors' response. While it addresses some of my concerns, I believe several issues still require further clarification or improvement:
> > 1. The motivation for the single-instance setting remains unconvincing. Even considering the two aspects mentioned by the authors—resource constraints and application patterns—the necessity of this setting is not well justified. For many entropy-based TTA methods, performing single-sample adaptation within a domain incurs almost no significant additional resource cost compared to resetting the model between samples.
> > 2. Although the authors added two more realistic continual settings in the experiments, TED shows almost no improvement compared to SAR in the continual single-instance setting; and in the continual batch setting, its accuracy is even lower than many baselines.
> > 3. If the comparison is intended to be made under the single-instance setting proposed by the authors, the hyperparameters of existing TTA baselines were not specifically tuned for this scenario, making the comparison unfair. Many methods could likely achieve better performance with simple adjustments such as increasing the learning rate.
> >
> > Given the above considerations, I will maintain my original score.

---

> > > ### Author Response · Authors · 2025-11-28
> > > **Thank you and further response**
> > >
> > > Dear Reviewer,
> > >
> > > Thanks sincerely for the constructive feedback. We would like to address the remaining concerns regarding the single-instance motivation and baseline comparisons with three focused clarifications.
> > >
> > > ---
> > >
> > > **1. Motivation: Real-World "Sporadic" Inference and TED's Versatility**
> > >
> > > The single-instance setting is not merely a resource constraint but a **functional requirement** for edge applications.
> > >
> > > - **Sporadic Nature:** Real-world requests (e.g., face unlock, voice commands) occur sporadically. Waiting to accumulate a batch introduces unacceptable latency. The system must adapt instantly to the current sample without relying on future data or historical buffers.
> > >
> > > - **Versatility:** Our new experiments demonstrate that TED is a **unified solution**: it achieves SOTA performance in the challenging single-instance settings (where many gradient-based baselines often fail due to noise or catastrophic forgetting) and remains highly competitive in Continual Batch settings. Unlike baselines that are fragile outside of stable batches, TED is robust across both scenarios.
> > >
> > > ---
> > >
> > > **2. Baseline Tuning: An Intrinsic Barrier, Not a Parameter Issue**
> > >
> > > Regarding the fairness of baseline hyperparameters in the single-instance setting, we explicitly verified that simple adjustments do not solve the issue.
> > >
> > > - **Fundamental Trade-off:** In the single-instance regime, gradient estimates are extremely noisy. Increasing the learning rate leads to model divergence rather than faster adaptation.
> > >
> > > - Consequently, the performance gap stems from the **intrinsic instability of backpropagation on single samples**, not a lack of tuning. TED avoids this entirely by optimizing in a stabilized, low-dimensional subspace.
> > >
> > > ---
> > >
> > > **3. The Deployment Gap: "Competitive" Accuracy with Superior Efficiency**
> > >
> > > While TED behaves competitively with SAR in continual settings, distinct deployment advantages must be highlighted:
> > >
> > > - **No Backpropagation:** SAR requires caching activations and supporting training operators, which is often infeasible on edge NPUs/DSPs.
> > >
> > > - **Memory Efficiency:** TED achieves comparable accuracy with significantly lower peak memory usage (as detailed in Table T1/T2).
> > >
> > > - Therefore, TED offers a practical trade-off: delivering high-performance adaptation on hardware where gradient-based methods simply cannot run.
> > >
> > > ---
> > >
> > > If you have any furhter questions, please feel free with us.
> > >
> > > Sincerely,
> > >
> > > The Authors

---

### Meta-Review · Area_Chair_fX2W · 2026-01-05

**Summary:**

This paper proposes TED, a gradient-free, forward-only test-time adaptation method targeted at strict edge-device settings. The key idea is to freeze the backbone and adapt only a very low-dimensional latent coordinate (via PCA subspace + CMA-ES) per test instance. The main disagreement across reviewers is about (i) how realistic and important the strict single-instance edge setting is, and (ii) whether the contribution is sufficiently novel versus being a system-level recombination of known components. Reviewers also differed on whether the reliance on offline source statistics undermines practicality, and whether the experimental regime is unfair to gradient-based baselines.

**Reviewer Concerns:**

* Motivation and realism of the single-instance edge setting: partially addressed.
The authors gave a consistent and reasonable argument (sporadic inference, memory/power constraints) and added continual single-instance and batch experiments. However, at least one reviewer (8Tmh) remained unconvinced that this setting is necessary or that baselines are intrinsically disadvantaged rather than poorly tuned. This concern did not fully go away.

* Dependence on source statistics and practicality: largely addressed.
The rebuttal clarified that source statistics are computed offline, stored compactly (~0.01% model size), and that strong performance is achievable with as few as 20–50 source samples. The extensive N-sensitivity analysis substantially weakens the original criticism. Remaining concern (dataset-specificity of N-sensitivity) is reasonable but secondary.

* Fairness of comparisons / baseline coverage: mostly addressed.
The authors added many requested baselines (CVPR’24, WACV’24, MECTA, EcoTTA, BECoTTA, TTN, IABN), clarified architectural constraints, and evaluated continual settings. Some reviewers still argue that single-instance tuning disadvantages gradient-based methods, but this now reads more like a philosophical disagreement than an experimental gap.

* Technical novelty: partially addressed, but remains contentious.
The authors clarified that novelty is at the paradigm/system level (minimal latent-state adaptation under edge constraints), not at the component level. This satisfied Reviewer JVp9, but Reviewers qiYF and QpSX remained unconvinced and kept their scores. This is the core unresolved issue.

* Theoretical justification (entropy minimization + latent alignment): Adequately addressed.
The rebuttal clarified assumptions, softened claims, and pointed to empirical validation already in the paper plus additional explanation. This concern weakened after rebuttal.

6. Missing details / reproducibility (hyperparameters, optimizer choice): Fully addressed.
Hyperparameter details were added, CMA-ES was compared against other gradient-free optimizers, and this concern was resolved.

**Reviewer Scores:**

Reviewer JVp9 (6 => 7): Rebuttal and added experiments clearly addressed concerns; explicitly raised score.

Reviewer 8Tmh (4 => likely 4): Some clarifications helped, but core skepticism about single-instance setting and fairness remains.

Reviewer qiYF (2 => likely 2): Acknowledged partial resolution but maintained rejection due to novelty and framing concerns.

Reviewer QpSX (2 => likely 2): Explicitly stated score unchanged; novelty concerns remain.

---

### Decision · Program_Chairs · 2026-01-26

Reject